# Locality Sensitive Hashing in Fourier Frequency Domain For Soft Set Containment Search

**Indradyumna Roy**[†]    **Rishi Agarwal**[†]
**Soumen Chakrabarti**[†]    **Anirban Dasgupta**[◇]    **Abir De**[†]
[†]IIT Bombay, [◇]IIT Gandhinagar
{indraroy15, rishiagarwal18, soumen, abir}@cse.iitb.ac.in
anirbandg@cse.iitgn.ac.in

## Abstract

In many search applications related to passage retrieval, text entailment, and subgraph search, the query and each 'document' is a set of elements, with a document being relevant if it contains the query. These elements are not represented by atomic IDs, but by embedded representations, thereby extending set containment to *soft* set containment. Recent applications address soft set containment by encoding sets into fixed-size vectors and checking for elementwise *vector dominance*. This 0/1 property can be relaxed to an asymmetric *hinge distance* for scoring and ranking candidate documents. Here we focus on data-sensitive, trainable indices for fast retrieval of relevant documents. Existing LSH methods are designed for mostly symmetric or few simple asymmetric distance functions, which are not suitable for hinge distance. Instead, we transform hinge distance into a proposed *dominance similarity* measure, to which we then apply a Fourier transform, thereby expressing dominance similarity as an expectation of inner products of functions in the frequency domain. Next, we approximate the expectation with an importance-sampled estimate. The overall consequence is that now we can use a traditional LSH, but in the frequency domain. To ensure that the LSH uses hash bits efficiently, we learn hash functions that are sensitive to both corpus and query distributions, mapped to the frequency domain. Our experiments show that the proposed asymmetric dominance similarity is critical to the targeted applications, and that our LSH, which we call FOURIERHASHNET, provides a better query time vs. retrieval quality trade-off, compared to several baselines. Both the Fourier transform and the trainable hash codes contribute to performance gains.

## 1   Introduction

Consider a corpus $X$ of sets $x$ (which we call 'documents') over some universe of discrete items, and let $q$ be a query which is also a subset of this universe. We wish to retrieve those $x \in X$ which satisfy $q \subseteq x$. In most real-world applications, the items in the universe are not just opaque IDs, but are embedded in a rich feature space, demanding that the definition of "$q \subseteq x$" be generalized suitably.

We formalize the notion of *soft set containment* by writing $q = \{q_i\}$ and $x = \{x_i\}$ and the corresponding sets of item embeddings as $\{\vec{q_i}\}$ and $\{\vec{x_i}\}$. If $q, x$ are sentences, $\vec{q_i}, \vec{x_i}$ may be per-word contextual embeddings output from a transformer. If $q, x$ are graphs, $\vec{q_i}, \vec{x_i}$ may be contextual node embeddings, such as those output by a Graph Neural Network (GNN). These set-of-vector representations of $q$ and $x$ are generally of variable sizes. A suitable set encoding gadget, such as simple pooling [41, 31] or a trainable Deep Set [60] or Set Transformer [27] network, converts them to fixed-size vectors given by $\boldsymbol{q} = \text{SetEnc}(\{\vec{q_i}\})$ and $\boldsymbol{x} = \text{SetEnc}(\{\vec{x_i}\})$, with $\boldsymbol{x}, \boldsymbol{q} \in \mathbb{R}^K$. Several applications [52, 26, 10, 31] then use the test "$\boldsymbol{q} \leq \boldsymbol{x}$" (elementwise vector dominance) as a surrogate for testing if $q \subseteq x$.

37th Conference on Neural Information Processing Systems (NeurIPS 2023).

To convert the Boolean test for vector dominance, $\boldsymbol{q} \leq \boldsymbol{x}$, into a graded score suitable for ranking (and backpropagation), these applications [52, 26, 10, 31] use a form of (asymmetric) **hinge distance**

$$d(q, x) = \left\| [\boldsymbol{q} - \boldsymbol{x}]_+ \right\|_1 = \sum_k \max\{0, \boldsymbol{q}[k] - \boldsymbol{x}[k]\}. \tag{1}$$

$d(q, x) = 0$ when $\boldsymbol{q} \leq \boldsymbol{x}$ holds elementwise, and measures the extent of the constraint violation otherwise. A search system must retrieve the top-$\tau$ documents $x$ with the smallest $d(q, x)$, given query $q$. Several example applications that fit into this framework are elaborated in Appendix B. Even if an application does not fit (1) exactly, our technique may help address other asymmetric distances.

**Our goal**   When corpus $X$ is large, it is impractical to evaluate (1) for each document $x$. Our goal is to retrieve these $\tau$ documents without explicitly evaluating $d(q, x)$ for all $x \in X$, within query time that scales slowly with $|X|$. To achieve this, we design an asymmetric Locality Sensitive Hashing (ALSH) method tailored for hinge distance (1), which then immediately addresses soft set-containment based search.

**Prior work and their limitations**   When set elements are represented by atomic IDs, Bloom filters [36] and maximum inner product search (MIPS) can be used to find the best $\tau$ corpus items that are closest to being supersets [46, 59, 45, 2]. However, these techniques are designed specifically for items with opaque IDs, rather than contextual embeddings. LSH [7, 53, 17, 19, 1] has been established as a standard technique for fast approximate near-neighbor search (e.g., FAISS, DPR) in the space of contextual embeddings. However, they predominantly work for symmetric notions of relevance, such as Jaccard similarity, dot product, cosine similarity, or Hamming distance, rather than asymmetric distances like (1). Neyshabur and Srebro [35] propose a LSH suited for asymmetric relevance (ALSH), but it does not provide a satisfactory solution for (1), as our experiments show.

## 1.1   Our contributions

Responding to the above motivations, we present FOURIERHASHNET, a new LSH for hinge distance-based asymmetric distance measures. Specifically, we make the following contributions.

**Scalable hinge distance search for soft set containment**   From several applications, we distil the strongly-motivated problem of fast top-$\tau$ retrieval using hinge distance (1), to capture soft set containment. To our knowledge, (A)LSH for hinge distance has not been explored till date.

**Transformation of hinge distance to enable ALSH design**   One could leverage its shift-invariant property to apply a Fourier transform on the *negative* distance, express it as the dot product similarity between the corresponding Fourier features and then use Asymmetric LSH (ALSH) [35]. However, as we show in Section 3.1, using the negative distance leads to singularities of the underlying Fourier transform at some points. This in turn does not allow us to design an LSH for such measure. We circumvent this problem by a suitable transformation of hinge distance to a **dominance similarity**, whose Fourier transform is absolutely convergent.

**Design of Fourier features**   Next, we propose a novel method of lifting the dense vectors to frequency domain, such that the dominance similarity in the original space can be expressed as the cosine similarity between the infinite dimensional Fourier features. However, our dominance similarity function is *not* a positive definite kernel. Hence, unlike Rahimi and Recht [40], we cannot apply Bochner theorem [44] to obtain finite dimensional Fourier features. Instead, we first scale the Fourier features with a sinc function and then obtain finite dimensional Fourier features via importance sampling.

**Trainable hashcode design**   The cosine similarity between the sampled Fourier features is the unbiased estimate of our dominance similarity measure. This allows the use of conventional random hyperplane LSH. However, such an LSH is not guided by the underlying data distribution. To mitigate this limitation, we compute the hashcodes by feeding the Fourier features into a trainable neural hashing network. Prior approaches [54, 15] to trainable hashing encourage bucket balance over the entire corpus, regardless of the query workload. However, this approach is not optimal if most corpus items are irrelevant for most queries, as is usually the case. We propose a new loss function that encourages the best-match hash bucket for a query to include relevant documents and exclude irrelevant documents.

**Experiments**   We show, through extensive experiments, that FOURIERHASHNET is more effective than existing LSH schemes, and that both frequency domain representations and the new trainable hashcode contribute to our gains.

## 2 Preliminaries

**Notation**  Throughout, we will use $[K]$ to mean $\{1, \ldots, K\}$ or $\{0, \ldots, K-1\}$ as convenient. We use $q$ to indicate a query and $x$ to indicate a corpus 'document'. Their (possibly learnt) representations are denoted by $\boldsymbol{x}, \boldsymbol{q} \in \mathbb{R}^K$. For supervision, $(q, x)$ may come with a binary relevance judgment $\mathrm{rel}(q, x) \in \{0, 1\}$. We have defined a potentially learnable distance $d(q, x)$ — a computable surrogate for $\mathrm{rel}(q, x)$ — above in Eqn. (1). One can define a similarity measure $\mathrm{sim}(q, x)$ by applying a monotonically decreasing function on the distance $d(q, x)$. We define $\iota = \sqrt{-1}$ and denote the set of corpus items as $X = \{x_1, x_2, ..., x_N\}$. We indicate the domain of query and corpus items as $\mathcal{Q}$ and $\mathcal{X}$ respectively. Given a function $s(t)$, its Fourier transform is the function $S : \mathbb{C} \to \mathbb{C}$ which satisfies $s(t) = \int_{-\infty}^{\infty} S(\iota\omega)e^{\iota\omega t}d\omega$, where $\omega$ is the frequency variable and $S(\iota\omega) = \frac{1}{2\pi}\int_{-\infty}^{\infty} s(t)e^{-\iota\omega t}dt$. For a vector $\boldsymbol{q}$ or $\boldsymbol{x} \in \mathbb{R}^K$, the Fourier transform is synthesized using a frequency vector $\boldsymbol{\omega} \in \mathbb{R}^K$ of same dimension as $\boldsymbol{x}$ or $\boldsymbol{q}$. Here, a function $s(\boldsymbol{x})$ can be expanded as $s(\boldsymbol{x}) = \int_{\boldsymbol{\omega} \in \mathbb{R}^K} S(\iota\boldsymbol{\omega})e^{-\iota\boldsymbol{\omega}^\top \boldsymbol{x}}d\boldsymbol{\omega}$.

### 2.1 Locality sensitive hashing

**Indexing corpus items**  Given a set of corpus items $X = \{x_1, x_2, ..., x_N\}$, an LSH will hash each item $x_i$, $L$ times, which is called the number of *trials*. For each trial $\ell \in [L]$, it prepares $B$ *buckets*, which are indexed as the pair $(\ell, b)$ with $\ell \in [L]$ and $b \in [B]$. In the context of LSH, we draw $L$ independent samples of hash functions $h^{(\ell)}$ from a single hash family $\mathcal{H}$, such that $h^{(\ell)} : \mathbb{R}^K \to [B]$. A corpus item $x$ is inserted in the bucket indexed $(\ell, h^{(\ell)}(\boldsymbol{x}))$, for each $\ell \in [L]$.

**Symmetric LSH**  Given a query $q$, a symmetric LSH computes bucket indices $(\ell, h^{(\ell)}(\boldsymbol{q}))$ for all $L$ using the *same* hash functions $h^{(\ell)}$ used for indexing the corpus. Only those items $x$ that are in bucket $(\ell, h^{(\ell)}(\boldsymbol{q}))$ are considered as *candidates*; overall, the candidates are in the union of these buckets. In the rest of the paper, we will describe retrieval for one bucket under one trial, with the understanding that $L$ buckets will contribute candidates. An LSH exists if the query and corpus items are hashed in the same bucket with high (low) probability as long as their similarities are high (low). Formally, we define symmetric LSH as follows.

**Definition 2.1** (Symmetric Locality Sensitive Hashing (LSH)).  Given a domain of queries $\mathcal{Q}$ and corpus $\mathcal{X}$ with $\mathcal{Q}, \mathcal{X} \subset \mathcal{Z}$ and a similarity measure $\mathrm{sim} : \mathcal{Z} \times \mathcal{Z} \to \mathbb{R}$. A distribution over mappings $\mathcal{H} : \mathcal{Z} \to \mathbb{N}$ is said to be a $(S_0, cS_0, p_1, p_2)$-LSH for the similarity function $\mathrm{sim}(\cdot)$ if for all $q \in \mathcal{Q}$ and $x \in \mathcal{X}$ we have, with $p_1 > p_2$ and $c < 1$,

- if $\mathrm{sim}(q, x) \geq S_0$, then $\Pr_{h \sim \mathcal{H}}[h(q) = h(x)] \geq p_1$
- if $\mathrm{sim}(q, x) \leq cS_0$, then $\Pr_{h \sim \mathcal{H}}[h(q) = h(x)] \leq p_2$.

The hash family $\mathcal{H}$ is tailored to the specific choice of similarity function $\mathrm{sim}(q, x)$ (equivalently, the distance $d(q, x)$). When $\boldsymbol{q}, \boldsymbol{x} \in \mathbb{R}^K$ and $\mathrm{sim}(q, x) = \cos(\boldsymbol{q}, \boldsymbol{x})$, the choice of $\mathcal{H}$ corresponds to the uncountable set of all hyperplanes in $K$ dimensions passing through the origin [9]. When $\mathrm{sim}(q, x)$ is the Jaccard similarity $|q \cap x|/|q \cup x|$, $\mathcal{H}$ is the space of *minwise independent* hash functions [7].

**Asymmetric LSH (ALSH)**  In many applications, like the current setup (1), we have asymmetric similarity where $\mathrm{sim}(q, x) \neq \mathrm{sim}(x, q)$. In such cases, we employ two different hash families $\mathcal{G}$ and $\mathcal{H}$ to determine the bucket of query and corpus respectively. Formally, we define ALSH as follows:

**Definition 2.2** (Asymmetric Locality Sensitive Hashing (ALSH) [35]).  An asymmetric LSH is $(S_0, cS_0, p_1, p_2)$-ALSH for a similarity function $\mathrm{sim}(\bullet, \bullet)$ over $\mathcal{Q}, \mathcal{X}$ if we have two different distributions over mappings $\mathcal{G}$ and $\mathcal{H}$ such that, with $p_1 > p_2$ and $c < 1$,

- if $\mathrm{sim}(q, x) \geq S_0$ then $\Pr_{g \sim \mathcal{G}, h \sim \mathcal{H}}[g(q) = h(x)] \geq p_1$
- if $\mathrm{sim}(q, x) \leq cS_0$ then $\Pr_{g \sim \mathcal{G}, h \sim \mathcal{H}}[g(q) = h(x)] \leq p_2$.

As an example, given $\|\boldsymbol{x}\| \leq 1$, consider $\mathrm{sim}(q, x) = \boldsymbol{q}^\top \boldsymbol{x}/\|\boldsymbol{q}\|_2$, which can be re-written as $\cos(\alpha(\boldsymbol{q}), \beta(\boldsymbol{x}))$, where $\alpha(\boldsymbol{q}) = [0; \boldsymbol{q}/\|\boldsymbol{q}\|_2], \beta(\boldsymbol{x}) = [\sqrt{1 - \|\boldsymbol{x}\|_2^2}; \boldsymbol{x}]$. Thus, we can apply random hyperplane hash on both $\alpha(\boldsymbol{x})$ and $\beta(\boldsymbol{x})$ to construct $g(q) = \mathrm{sign}(\mathbf{w} \cdot \alpha(\boldsymbol{q}))$ and $h(x) = \mathrm{sign}(\mathbf{w} \cdot \beta(\boldsymbol{x}))$ with $\mathbf{w} \sim \mathcal{N}(\mathbf{0}, \boldsymbol{I})$. If $\|\boldsymbol{x}\|$ is unbounded, no ALSH exists for $\mathrm{sim}(q, x) = \boldsymbol{q}^\top \boldsymbol{x}/\|\boldsymbol{q}\|_2$ [35]. In $(S_0, cS_0, p_1, p_2)$-ALSH, retrieval of items with similarity score more than $S_0$ out of a database of items having a similarity score less than $cS_0$ will admit time-complexity $O(n^\rho \log n)$ and space complexity $O(n^{1+\rho})$ where $\rho = \log p_1 / \log p_2$ [35].

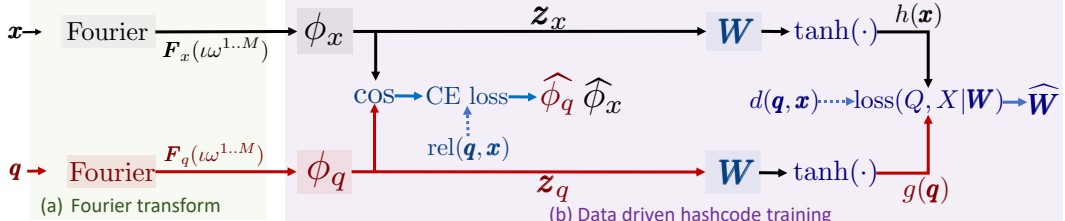

Figure 1: Outline of FOURIERHASHNET. **(a)** Given the input query embedding $q$ and corpus embedding $x$, we apply the asymmetric transformation in Eq. (10) to obtain corresponding Fourier features $F_q(\iota\omega^{1..M})$ and $F_x(\iota\omega^{1..M})$. **(b)** We use the generated Fourier features as inputs, to train asymmetric Fourier transformation networks $\phi_q$ and $\phi_x$ using Eq. (12). This generates transformed Fourier representations $z_q = \phi_q(F_q(\iota\omega^{1..M}))$ and $z_x = \phi_x(F_x(\iota\omega^{1..M}))$, which are in turn used to train the random hyperplanes $W$ using Eq. (14). The trained $\widehat{\phi_q}, \widehat{\phi_x}$ and $\widehat{W}$ thus obtained are used to generate final hashcodes $g(q) = \text{sign}(\widehat{W}\widehat{\phi_q}(F_q(\iota\omega^{1..M})))$, $h(x) = \text{sign}(\widehat{W}\widehat{\phi_x}(F_x(\iota\omega^{1..M})))$.

## 2.2 Problem statement

Given the set of training queries $Q$ and corpus $X$, with supervised relevance scores $\text{rel}(q, x) \in \{0, 1\}$ and the surrogate score $d(q, x)$ defined in Eq. (1), we aim to design an LSH of the distance $d(q, x)$ which can efficiently retrieve top-$\tau$ corpus items for any new query $q'$.

**Why are existing methods not suitable?**   As we discussed in Section 2.1, relevance metrics for popular LSHs are mostly symmetric, *e.g.*, cosine, dot-product, and Jaccard similarity. In particular, Jaccard similarity, although commonly used in set-related applications, is not suitable for our problem, where we define $\text{rel}(q, x) = 1$ when $q \subseteq x$ and 0 otherwise — it is possible that there exists a higher overlap between $q$ and $x$ when $q \not\subseteq x$, and a lower overlap when $q \subseteq x$. E.g., suppose $q = \{a, b\}$, $x_1 = \{a, b, c, d, e\}$, and $x_2 = \{b\}$. Here, $\text{rel}(q, x_1) = 1$ and $\text{rel}(q, x_2) = 0$. However, the Jaccard similarity $J(q, x)$ is not able to reflect the order of $\text{rel}(q, x)$ since $J(q, x_1) = 2/5 < J(q, x_2) = 1/2$.

As discovered by Charikar [9, Lemma 1], the similarity functions in symmetric LSH are inversely related to a *metric*, which must satisfy symmetry and triangle inequality. Although a query normalized dot product similarity appears asymmetric, it can be expressed using cosine similarity. This readily allows us to use a random hyperplane based (asymmetric) LSH. In contrast, it is not immediately apparent how to find such a connection for our asymmetric hinge distance (1).

## 3 FOURIERHASHNET: A new ALSH for hinge distance search

**Overview of our approach**   We design an ALSH for $d(q, x)$ in three steps. In the first step, we construct a suitable dominance similarity function $\text{sim}(q, x)$ from $d(q, x)$ in such a way that there exists a probability distribution $p : \mathbb{R}^K \to [0, 1]$ and bounded Fourier representations $F_q(\iota\omega)$ and $F_x(\iota\omega)$ of both query $q$ and corpus items $x$ such that

$$\text{sim}(q, x) = \int_{\omega \in \mathbb{R}^K} F_q(\iota\omega)^\top F_x(\iota\omega) p(\omega) d\omega = \mathbb{E}_{\omega \sim p(\bullet)}[F_q(\iota\omega)^\top F_x(\iota\omega)] \tag{2}$$

In the second step, we approximate the expected value of the $F_q(\iota\omega)^\top F_x(\iota\omega)$ using a finite sample of Fourier features. This allows us to apply random hyperplane LSH , similar to asymmetric dot product LSH. However, these hyperplanes are drawn from an isotropic Gaussian distribution in a data-oblivious manner, which results in suboptimal bucket distribution in terms of accuracy-efficiency trade off. To tackle this issue, in the third step, we train the random hyperplanes $W$ which takes the Fourier features as input and give (soft) binary hashcodes, which are optimized to effectively trade off between accuracy and efficiency. Next, we provide the details of the above three steps.

### 3.1 Design of dominance similarity function $\text{sim}(q, x)$ from hinge distance

**Limitations of simple choices of dominance similarity function** $\text{sim}(q, x)$   A dominance similarity function $\text{sim}(q, x)$ is inversely related to the hinge distance $d(q, x)$. Chierichetti and Kumar [11] characterized that, any function of a similarity measure is LSHable, if and only if this function is a probability generating function. However, this characterization applies only to symmetric LSH and no such guiding principle is available for an ALSH. In this context, one can experiment with simple designs of sim that are inversely related to $d$. An immediate choice is $\text{sim}(q, x) = -d(q, x)$.

However, if we allow $q$ and $x$ to be any vector from $\mathbb{R}^K$, then $\text{sim}(q, x)$ is not bounded. Finding ALSH for unbounded similarity measures is extremely difficult if not impossible. For example, no ALSH exists even for dot product similarity in two or more dimensions [35]. Moreover, suppose we express $\text{sim}(q, x)$ using the Fourier expansion

$$\text{sim}(q, x) = \sum_{k \in [K]} \int_{-\infty}^{\infty} S(\iota\omega_k) e^{\iota\omega_k(\bm{q}[k] - \bm{x}[k])} d\omega_k. \tag{3}$$

Then, $S(\iota\omega)$, *i.e.*, the Fourier transform of the function $s(t) = -[t]_+$ used in each dimension, has a singularity at $\omega = 0$. In particular, we have $S(\iota\omega) = -\iota\delta'(\omega)/2 + 1/2\pi\omega^2$. Here, $\delta'(\omega)$ is the derivative of Dirac delta functional. Thus, $S(\iota\omega)$ becomes unbounded as $\omega \to 0$. These issues eventually prevent us from designing bounded Fourier features $\bm{F}_q(\iota\bm{\omega})$ and $\bm{F}_x(\iota\bm{\omega})$ for Eq. (2).

$\text{sim}(q, x)$ **with bounded Fourier transform**    The key reason for which $S(\iota\omega)$ becomes unbounded as $\omega \to 0$ is that the function $s(t) = -[t]_+$ is unbounded at $t \to \infty$. However, in practice, the embeddings are bounded and we have a bounded difference $|\bm{q}[k] - \bm{x}[k]| \leq T$. Thus, it is reasonable to ignore the effect of $[\bm{q}[k] - \bm{x}[k]]_+$ when $|\bm{q}[k] - \bm{x}[k]| > T$. To this end, we compute $\text{sim}(q, x)$ as

$$\text{sim}(q, x) = \sum_{k \in [K]} s(\bm{q}[k] - \bm{x}[k]), \qquad \text{where } s(t) = \begin{cases} T - t & \text{if } 0 \leq t \leq T, \\ T & \text{if } -T \leq t < 0, \\ 0 & \text{otherwise.} \end{cases} \tag{4}$$

In practice, we choose $T$ as a hyperparameter greater than $\max_k |\bm{q}[k] - \bm{x}[k]|$. Upon restricting the computation within this domain, one can immediately show that $\text{sim}(q, x) = KT - d(q, x)$.

### 3.2 Computation of finite dimensional Fourier features for dominance similarity $\text{sim}(q, x)$

**Fourier transform of $s(t)$**    We next compute the Fourier representation $S(\iota\omega)$ of $s : \mathbb{R} \to \mathbb{R}$ (4).

**Proposition 3.1.** (Proven in Appendix D) $s(t)$ specified in Eq. (4) has Fourier transform

$$S(\iota\omega) = \underbrace{\frac{T\sin(\omega T)}{2\pi\omega} + \frac{\sin^2(\frac{\omega T}{2})}{\pi\omega^2}}_{\text{Re}(S(\iota\omega))} + \iota \underbrace{\left[ \frac{\sin(\omega T)}{2\pi\omega^2} - \frac{T\cos(\omega T)}{2\pi\omega} \right]}_{\text{Im}(S(\iota\omega))} \tag{5}$$

While the Fourier transform of $-[t]_+$ is unbounded as $\omega \to 0$, here, $S(\iota\omega)$ is bounded everywhere.

**Computation and sampling of Fourier features**    Once we compute $S(\iota\omega)$ using Eq. (5), we use Eq. (3) to compute $\text{sim}(q, x)$ as follows:

$$\text{sim}(q, x) = \sum_{k \in [K]} \int_{-\infty}^{\infty} [\text{Re}(S(\iota\omega_k)) + \iota\text{Im}(S(\iota\omega_k))] e^{\iota\omega_k(\bm{q}[k] - \bm{x}[k])} d\omega_k \tag{6}$$

Now, we define $\bm{S}_q(\iota\bm{\omega})$ and $\bm{S}_x(\iota\bm{\omega})$, the query and corpus specific Fourier representations:

$$\bm{S}_q(\iota\omega_k) = \Big[ u_k \sqrt{|\text{Re}(S(\iota\omega_k))|} \big[ \cos(\omega_k\bm{q}[k]), \sin(\omega_k\bm{q}[k]) \big],$$
$$v_k \sqrt{|\text{Im}(S(\iota\omega_k))|} \big[ -\sin(\omega_k\bm{q}[k]), \cos(\omega_k\bm{q}[k]) \big] \Big]$$
$$\bm{S}_x(\iota\omega_k) = \Big[ \sqrt{|\text{Re}(S(\iota\omega_k))|} \big[ \cos(\omega_k\bm{x}[k]), \sin(\omega_k\bm{x}[k]) \big],$$
$$\sqrt{|\text{Im}(S(\iota\omega_k))|} \big[ \cos(\omega_k\bm{x}[k]), \sin(\omega_k\bm{x}[k]) \big] \Big] \tag{7}$$

Here, $u_k = \text{sign}(\text{Re}(S(\iota\omega_k)))$, $v_k = \text{sign}(\text{Im}(S(\iota\omega_k)))$. They ensure that the dot-product $\bm{S}_q(\iota\omega_k)^\top \bm{S}_x(\iota\omega_k)$ equals to the real part of the integrand in the RHS of Eq. (6). Since the dominance similarity $\text{sim}(q, x)$ is a real quantity, the imaginary part of the RHS integrates to zero. Therefore, using the dot product of the vectors $\bm{S}_q(\iota\omega_k)$ and $\bm{S}_x(\iota\omega_k)$, which are purely real, we can express

$$\text{sim}(q, x) = \int_{-\infty}^{\infty} \sum_{k \in [K]} \bm{S}_q(\iota\omega_k)^\top \bm{S}_x(\iota\omega_k) d\omega_k = \int_{\bm{\omega} \in \mathbb{R}^K} \bm{S}_q(\iota\bm{\omega})^\top \bm{S}_x(\iota\bm{\omega}) d\bm{\omega} \tag{8}$$

Here, $\bm{S}_\bullet(\iota\bm{\omega}) = [\bm{S}_\bullet(\iota\omega_1), .., \bm{S}_\bullet(\iota\omega_K)]$, $\bm{\omega} = [\omega_1, .., \omega_K]$. Note that, the expression of $\bm{S}_q(\iota\omega_k)$ is different from $\bm{S}_x(\iota\omega_k)$ in Eq. (7). This maintains the asymmetry in the final dot product in Eq. (8).

Inspired by the seminal work of Rahimi and Recht [40], several works [50, 37] have exploited Fourier transformations to approximate various functions using inner product between the feature maps. However, the functions that Rahimi and Recht [40] considered are shift invariant positive definite kernels. This allowed them to leverage Bochner's theorem [44] which establishes that the Fourier

transformation of these kernels are probability distributions. However, in Eq. (8), there is no such readily available probability distribution. In response, we attempt to find out a probability distribution $p(\boldsymbol{\omega})$ which allows us to draw samples using an importance sampling like procedure, as follows:

$$\text{sim}(q, x) = \mathbb{E}_{\boldsymbol{\omega} \sim p(\boldsymbol{\omega})} \left[ \boldsymbol{F}_q(\iota\boldsymbol{\omega})^\top \boldsymbol{F}_x(\iota\boldsymbol{\omega}) \right], \text{ where, } \boldsymbol{F}_q(\iota\boldsymbol{\omega}) = \frac{\boldsymbol{S}_q(\iota\boldsymbol{\omega})}{\sqrt{p(\boldsymbol{\omega})}}, \boldsymbol{F}_x(\iota\boldsymbol{\omega}) = \frac{\boldsymbol{S}_x(\iota\boldsymbol{\omega})}{\sqrt{p(\boldsymbol{\omega})}}, \quad (9)$$

Let $\{\boldsymbol{\omega}^j\}_{j=1}^M \sim p(\boldsymbol{\omega})$ be $M$ i.i.d random samples. We compute the Monte Carlo estimate as follows:

$$\text{sim}(q, x) \approx \frac{1}{M} \sum_{j \in [M]} \boldsymbol{F}_q(\iota\boldsymbol{\omega}^j)^\top \boldsymbol{F}_x(\iota\boldsymbol{\omega}^j) \propto \cos(\boldsymbol{F}_q(\iota\boldsymbol{\omega}^{1..M}), \boldsymbol{F}_x(\iota\boldsymbol{\omega}^{1..M})) \quad (10)$$

Here, $\boldsymbol{F}_\bullet(\iota\boldsymbol{\omega}^{1..M}) = [\boldsymbol{F}_\bullet(\iota\boldsymbol{\omega}^1), .., \boldsymbol{F}_\bullet(\iota\boldsymbol{\omega}^M)]$. Note that, as suggested by Eqs. (7) and (9), $||\boldsymbol{F}_q(\iota\boldsymbol{\omega}^{1...M})||_2 = ||\boldsymbol{F}_x(\iota\boldsymbol{\omega}^{1..M})||_2 = \sum_{j=1}^M \sum_{k=1}^K \frac{|\text{Re}(S(\iota\omega_k^j))| + |\text{Im}(S(\iota\omega_k^j))|}{p(\omega_k^j)}$. Thus, the value is independent of the query or corpus, which leads to the proportionality relation. We choose the probability distribution $p(\boldsymbol{\omega})$ guided the proportionality constant $||\boldsymbol{F}_\bullet(\iota\boldsymbol{\omega}^{1..M})||$ and set $p(\boldsymbol{\omega}) = \prod_{k \in [K]} p(\omega)$, where $p(\omega) \propto |\text{Re}(S(\iota\omega))| + |\text{Im}(S(\iota\omega))|$. However, the integral of these terms may not be bounded. Therefore, we set the support of $p(\omega)$ between $[-\omega_{\max}, \omega_{\max}]$, thus eliminating the higher frequency terms. The effect on the overall score is small. Attenuation of the higher frequency signals can be seen as a multiplication with a low pass filter in the frequency domain, which affects a convolution in the time domain. Its impact on the similarity score is proportional to $1/\omega_{\max}$. This still allows us for ALSH despite frequency truncation.

**Theorem 3.2.** (Proven in Appendix D) Let $q, x \in \mathbb{R}^K$, $\cos^{-1}$ be Lipschitz with Lipschitz constant $L_{\cos}$; the hyperparameter $T$ in Eq. (4) be chosen such that $T > ||q - x||_\infty$; the frequency sampling distribution $p(\omega_k^j) \propto [|\text{Re}(S(\omega_k^j))| + |\text{Im}(S(\omega_k^j))|]$ with the support set $\omega_k^j \in [-\omega_{\max}, \omega_{\max}]$ and the proportionality constant $I(\omega_{\max}) = \int_{-\omega_{\max}}^{\omega_{\max}} [|\text{Re}(S(\omega)| + |\text{Im}(S(\omega))|]d\omega$. Then, the mapping $g(q)[i] = \text{sign}(\mathbf{w}_i^\top \boldsymbol{F}_q(\boldsymbol{\omega}^{1...M}))$ and $h(x)[i] = \text{sign}(\mathbf{w}_i^\top \boldsymbol{F}_x(\boldsymbol{\omega}^{1...M}))$ where $\mathbf{w}_i \sim N(0, \mathbb{I})$, constitutes a $(S_0, cS_0, p_1, p_2)$-ALSH for some $p_1$ and $p_2$ if we choose the support set $[-\omega_{\max}, \omega_{\max}]$ and the number of samples $M$ as follows:

$$\omega_{\max} > \frac{4KL_{\cos}}{\pi(1-c)S_0} \left( 6 + \frac{2T}{T - \max_{x,q} ||x - q||_\infty} \right) \text{ and } M > \left[ \frac{4L_{\cos}}{(1-c)S_0} \right]^2 KI(\omega_{\max}). \quad (11)$$

Based on the outlined assumptions, the above Theorem guarantees that FOURIERHASHNET is an $(S_0, cS_0, p_1, p_2)$-ALSH for the asymmetric dominance similarity score, subject to appropriate choices of $\omega_{\max}$ to bound the effect of frequency truncation and $M$ to bound the variance of the Monte Carlo sample estimate.

### 3.3 Trainable hashing network

**Random hyperplane LSH** Eq. (10) provides an asymmetric transformation on the input query-corpus pair, which maps it into the cosine similarity space, thus allowing for Random Hyperplanes hashing. We sample $H$ spherically symmetrically distributed normal vectors $\{\mathbf{w}_i\}_{i=1}^H$, *i.e.*, $\mathbf{w}_i \sim \mathcal{N}(0, \mathbb{I})$, each perpendicular to a random hyperplane passing though the origin. For each query $q$ and the corpus $x$, we can generate $H$-bit hashcodes $g(q), h(x) \in \{\pm 1\}^H$ from the Fourier features (10) as follows: $g(q)[i] = \text{sign}(\mathbf{w}_i^\top \boldsymbol{F}_q(\iota\boldsymbol{\omega}^{1...M}))$ and $h(x)[i] = \text{sign}(\mathbf{w}_i^\top \boldsymbol{F}_x(\iota\boldsymbol{\omega}^{1...M}))$. Consequently, we can index the given corpus with $N$ items, into a hash table with $2^H$ buckets. For each query $q$, we restrict our search within bucket $b = g(q)$. If the corpus items are uniformly distributed across all buckets, then it enables sub-quadratic time retrieval with $N/2^H$ comparisons (per trial).

**Data driven hashcode generation** The above random hyperplane LSH approach suffers from two distinct limitations: (1) the quality of Monte Carlo approximation obtained in Eq. (10), depends on the suitability of $p(\boldsymbol{\omega})$, and (2) the hyperplanes are data oblivious. Data oblivious hyperplanes provide the best efficiency if the corpus embeddings are uniformly spread over the $K$ dimensional sphere, which allows the random hyperplanes to evenly allocate the corpus items across different hashcodes. However, in practice, the spatial distribution of the embeddings is not uniform. This results in a skewed distribution of the corpus items across the hash buckets.

To tackle the first problem, we improve the quality of the Fourier features through a trainable nonlinear transformation. Here, we use two networks $\phi_q$ and $\phi_x$ which takes the Fourier features for the query and corpus, *i.e.*, $\boldsymbol{F}_q(\iota\boldsymbol{\omega}^{1..M})$ and $\boldsymbol{F}_x(\iota\boldsymbol{\omega}^{1..M})$ as input and outputs corresponding

transformed Fourier representations $\boldsymbol{z}_q = \phi_q(\boldsymbol{F}_q(\iota\boldsymbol{\omega}^{1..M}))$ and $\boldsymbol{z}_x = \phi_x(\boldsymbol{F}_x(\iota\boldsymbol{\omega}^{1..M}))$. We train $\phi_q$ and $\phi_x$ by minimizing a BCE loss on $\{\cos(\boldsymbol{z}_q, \boldsymbol{z}_x), \mathrm{rel}(q, x)\}$ pairs for $q \in Q$ and $x \in X$ as follows:

$$\min_{\phi_q, \phi_x} \sum_{q\in Q, x\in X} -\big[\mathrm{rel}(q, x)\log(1 + \cos(\boldsymbol{z}_q, \boldsymbol{z}_x)) + (1 - \mathrm{rel}(q, x)\log(1 - \cos(\boldsymbol{z}_q, \boldsymbol{z}_x))\big] \quad (12)$$

Next, we train the random hyperplanes $\boldsymbol{W} = [\mathbf{w}_1, \mathbf{w}_2, ..]$ using the transformed Fourier features $\{\boldsymbol{z}_q\}$ and $\{\boldsymbol{z}_x\}$. The final hashcodes $g(q)$ and $h(x)$ are obtained as $g(q) = \mathrm{sign}(\widehat{\boldsymbol{W}}\boldsymbol{z}_q)$, $h(x) = \mathrm{sign}(\widehat{\boldsymbol{W}}\boldsymbol{z}_x)$, where $\widehat{\boldsymbol{W}}$ are the final trained random hyperplanes. For training purposes, we use $\tanh(\boldsymbol{W}\bullet)$ as a smooth surrogate of $\mathrm{sign}(\boldsymbol{W}\bullet)$. The loss function $\mathrm{loss}(Q, X \mid \boldsymbol{W})$ used to train $\boldsymbol{W}$ consists of three components.

**(1) Collision minimizer** For any query $q$, our goal is to ensure that assigned bucket contains only positive items. Assuming corpus items are uniformly distributed across buckets, we ensure that for any query $q$, the $N/2^H$ most relevant items $X_{q\checkmark}$ measured in terms of $d(q, x)$ will have higher amount of bit overlap than rest of the items $X_{q\boldsymbol{\chi}}$. Here, $X_{q\checkmark}$ and $X_{q\boldsymbol{\chi}}$ indicate positive and negative silver instances (not gold instances) indicating top $N/2^H$ items *in terms of the (possibly trained) hinge distance $d(q, x)$*. We encode this by minimizing the following ranking loss.

---

**Algorithm 1** FOURIERHASHNET

1: **function** Train($X$,$Q$, $\{\mathrm{rel}(q, x)\}_{q\in Q, x\in X}$)
2:     Draw $\boldsymbol{\omega}^{1...M} \sim p(\boldsymbol{\omega})$
3:     Compute $\boldsymbol{F}_q(\iota\boldsymbol{\omega}^{1..M})$, $\boldsymbol{F}_x(\iota\boldsymbol{\omega}^{1..M})$ (Eq. (9))
4:     Train $\phi_q, \phi_x$ from $\mathrm{rel}(q, x)$, $\boldsymbol{F}_\bullet(\iota\boldsymbol{\omega}^{1..M})$ (Eq. (12))
5:     Train $\boldsymbol{W}$ by minimizing the loss (14)
6:     **Return** $\widehat{\phi}_x, \widehat{\phi}_q, \widehat{\boldsymbol{W}}$

1: **function** Index($\{\boldsymbol{F}_x(\iota\boldsymbol{\omega}^{1..M})\}_{x\in X}$)
2:     **Require:** Trained networks $\widehat{\phi}_x, \widehat{\boldsymbol{W}}$
3:     $h(x) \leftarrow \mathrm{sign}(\widehat{\boldsymbol{W}}\widehat{\phi}_x(\boldsymbol{F}_x(\iota\boldsymbol{\omega}^{1..M}))) \,\forall x \in X$
4:     **for** $x \in X$ **do**
5:         hash $x$ to bucket $b = h(x)$
6:     **Return** the bucket sets $B$

1: **function** Retrieve($q'$)
2:     **Require:** Trained networks $\widehat{\phi}_q, \widehat{\boldsymbol{W}}$
3:     Compute $\boldsymbol{F}_q(\iota\boldsymbol{\omega}^{1...M})$ based on $q'$
4:     $g(q') \leftarrow \mathrm{sign}(\widehat{\boldsymbol{W}}\widehat{\phi}_q(\boldsymbol{F}_q(\iota\boldsymbol{\omega}^{1...M})))$
5:     Rank all $x$ in the bucket $b = g(q')$ based on the distance $d(q', x)$ to obtain the list $\mathrm{List}_{q'}$.
6:     **Return** $\mathrm{List}_{q'}$

---

$$\Delta_1 = \sum_{q\in Q} \sum_{x\in X_{q\checkmark}, x'\in X_{q\boldsymbol{\chi}}} \big[1 + \tanh(\boldsymbol{W}\boldsymbol{z}_q)^\top \tanh(\boldsymbol{W}\boldsymbol{z}_{x'}) - \tanh(\boldsymbol{W}\boldsymbol{z}_q)^\top \tanh(\boldsymbol{W}\boldsymbol{z}_x)\big]_+ \quad (13)$$

This loss encourages that $\tanh(\boldsymbol{W}\boldsymbol{z}_q)^\top \tanh(\boldsymbol{W}\boldsymbol{z}_x) > \tanh(\boldsymbol{W}\boldsymbol{z}_q)^\top \tanh(\boldsymbol{W}\boldsymbol{z}_{x'}) + 1$, *i.e.*, the number of common bits between $q$ and $x \in X_{q\checkmark}$ is atleast one more than the same between $q$ and $x'$.

**(2) Fence Sitting** We set fence sitting loss as $\Delta_2 = \sum_{x\in X} \||\tanh(\boldsymbol{W}\boldsymbol{z}_x)| - 1\|_1$. This prevents the optimizer from arriving at a trivial solution by setting all hashcodes to zero.

**(3) Bit Balance** We set the bit balance loss as $\Delta_3 = \sum_{i\in[H]} |\sum_{x\in X} \tanh(\boldsymbol{W}\boldsymbol{z}_x)[i]|$. This enforces that each position should have an equal number of $+1$ and $-1$, thus ensuring that each random hyperplane evenly splits the set of points. Finally, we estimate $\boldsymbol{W}$ by minimizing the loss, with $\lambda_\bullet$ as hyperparameters such that $\sum_i \lambda_i = 1$, which is given as follows:

$$\mathrm{loss}(Q, X \mid \boldsymbol{W}) = \lambda_1 \Delta_1 + \lambda_2 \Delta_2 + \lambda_3 \Delta_3, \quad (14)$$

Algorithm 1 summarizes the overall procedure.

**Difference from existing trainable LSH** LSH training has been extensively studied [54, 15, 43], with Fence Sitting and Bit Balance losses being well known. However, the Collision Minimizer loss differs significantly from existing approaches. Current techniques seek to ensure load balance across hash buckets for all corpus items, including the ones that may not be relevant to most queries. This is unnecessary for query workloads which touch upon only a small subset of the corpus to generate the best responses. In contrast, our Collision Minimizer loss ensures that only the top-most bucket for any given query allows relevant items and explicitly denies irrelevant items. Thus, it is informed by the query workload, rather than assuming load balance for all items in the corpus. Such an approach may result in balanced bucket loads, but not necessarily.

## 4 Experiments

In this section, we provide a comprehensive evaluation of our method against several baselines and ablations on four datasets. Appendix F describes additional experiments. Our code is in `https://github.com/structlearning/fhashnet`.

### 4.1 Experimental setup

**Datasets** We experiment on datasets sampled from anonymized real-world Web log data, *viz*, MSWEB and MSNBC. MSWEB [5] is generated using logs from *www.microsoft.com*, containing

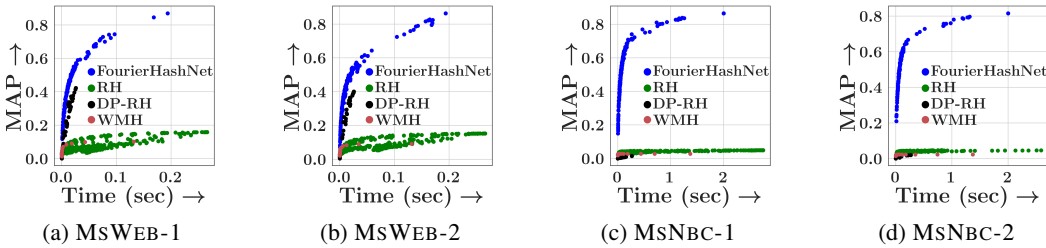

|   |   |   |   |
|---|---|---|---|
| (a) MsWeb-1 | (b) MsWeb-2 | (c) MsNbc-1 | (d) MsNbc-2 |

Figure 2: Effect of different similarity measures on LSH, measured in terms of variation of MAP vs. average query time (in sec) for all methods. Here, the final score used for ranking the relevant items is that similarity score for which the LSH is designed for.

records of the areas of the website visited by the users. MsNbc [8] is a collection of logs of user page requests from *msnbc.com*. In both cases, a record (either $q$ or $x$), is a passage that is regarded as a bag of words. Given a collection $V$ of such word bags, ($|V| = 11234$ for MsWeb and $|V| = 111290$ for MsNbc), we sample $|Q| = 500$ bags from $V$, designating them as queries, and designate the rest as corpus items $X = V \backslash Q$. Consistent with typical information retrieval application scenarios [51], we generate gold relevance labels based on (multi)set containment for MsWeb (MsNbc). (Additional methods for evaluation are explored in Appendix F.) We build the query set $Q$, such that the number of relevant items $N_{q\oplus} = |\{x \in X : \text{rel}(q, x) = +1\}| \in [5, 500]$ for each query $q$. We create four datasets by changing average relevance counts per query, $\overline{N}_{q\oplus}$. They are: (1) MsWeb-1 where $\overline{N}_{q\oplus} = 35.624$. (2) MsWeb-2 where $\overline{N}_{q\oplus} = 20.392$. (3) MsNbc-1 where $\overline{N}_{q\oplus} = 24.09$ (4) MsNbc-2 where $\overline{N}_{q\oplus} = 19.78$ The set of queries $Q$ is partitioned into 20% training set $Q_{\text{tr}}$, 20% validation set $Q_{\text{dev}}$ and 60% test set $Q_{\text{test}}$.

**Design of query and corpus embeddings $q, x$**  We begin with a pre-trained sentence transformer model [41] to obtain 768 dimensional dense contextual representations $\textbf{feature}_q$ and $\textbf{feature}_x$ for the each word in bags $q$ and $x$. Embeddings of words belonging to a bag are fed into a deep set [60] network to obtain a bag representation $q, x \in \mathbb{R}^K$, with $K = 294$ (chosen via hyperparameter sweep). To train the parameters inside the deep set network, we use $q, x$ to compute the proposed asymmetric hinge distance $d(q, x)$ (1), feed it into a trainable sigmoid layer $\sigma$ and minimize

$$\sum_{q,x} \text{BCE}\left(\text{rel}(q, x), \sigma(-d(q, x))\right) \tag{15}$$

which uses a BCE loss on the gold relevance labels. Once we obtain $q$ and $x$, we use Algorithm 1 to obtain trained $\widehat{\phi_q}, \widehat{\phi_x}$ and $\widehat{W}$ (Train($\cdot$)), which are then used for indexing (Index($\cdot$)).

**Evaluation**  Given a test query $q \in Q_{\text{test}}$ and a set of $N'_q$ candidate corpus items, we rank them in increasing order of their hinge distances $d(q, x)$ . Then we evaluate the average precision (AP) for the query and average over queries to report mean average precision (MAP) — see Appendix E.6.

## 4.2 Effect of different similarity measures on LSH

**Setup**  Here, we compare FourierHashNet against the three LSH baselines, *viz*, Random hyperlane (RH) [9], Dot product LSH (DP-RH) [35] and Weighted MinHash (WMH) [12], that are tailored towards cosine similarity, dot product similarity and Weighted Jaccard Similarity, respectively. For each LSH method, we train the embeddings $q, x$ and the final hashcodes $g(q)$ and $h(x)$ using the same networks, as in our method. Furthermore, we set the final relevance measure for ranking to be the similarity score for which the LSH is designed.

**Results**  We vary the mixing hyperparameters $\lambda_1$ and $\lambda_2$ in our loss (14) and the number of buckets $B$ to explore the tradeoff between accuracy (MAP) and average query time. In Figure 2, we summarize the results. We observe that: **(1)** FourierHashNet outperforms all the baselines by providing significantly better time-vs.-MAP trade-off across all datasets. In MsWeb datasets, all the baselines except DP-RH show poor performance. All baselines perform poorly for the MsNbc dataset. We remark that cosine similarity, dot product or weighted Jaccard similarity are not suited for vector dominance search. Therefore, the maximum possible MAP obtained by them are severely constrained. **(2)** In MsWeb datasets, DP-RH performs moderately, by achieving a MAP value around 0.4–0.42 within 0.03 seconds (average query time). This is because dot product can be computed significantly faster than all the other distance/similarity measures. In particular, it is $\sim 7.5 \times$ faster than our hinge distance (1), $\sim 10.3 \times$ faster than cosine, and $\sim 5.1 \times$ faster than Jaccard similarity.

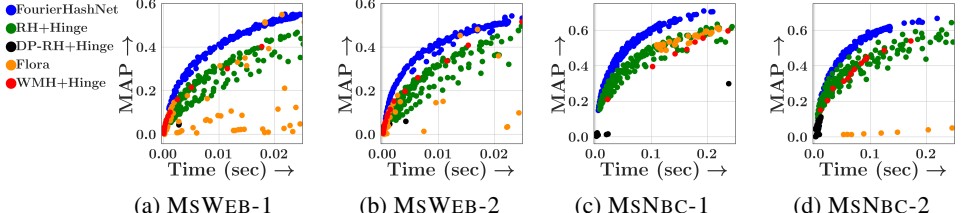

Figure 3: Trade-off between query time and accuracy (MAP) for MSWEB and MSNBC datasets where there is ≥10X speedup compared to exhaustive search. We apply different LSH methods on hinge distance guided embeddings, *viz*, RH+Hinge, DP-RH+Hinge, WMH+Hinge, FLORA and FOURIERHASHNET; and, then use the hinge distance to finally rank the retrieved items.

### 4.3 Comparison against other efficient indexing techniques

In Section 4.2, we used the similarity score corresponding to each LSH method for final candidate ranking. The baselines performed poorly, which may result from a poor choice of final similarity score or the indexing method. Here, we evaluate FOURIERHASHNET against baseline indexing methods applied on hinge distance guided embeddings. We consider LSH and Inverted File Indexing (IVF) based indexing variants in this section, and discuss graph-based indexing methods in Appendix F.

**Comparison with LSH based indexing** . Shrivastava and Li [47] showed that an LSH not tailored to the final scoring function may still provide an effective filter. Accordingly, we set the final similarity function to be dominance similarity, and compare against four possible LSH baselines.

Given the embeddings $q, x$ trained (15) using hinge distance, we feed them into the four baselines, each of which trains a hashing network in a different way. **(1)** RH+Hinge: We train a set of random hyperplanes represented by $W$ and compute the hashcodes as $h(q) = \text{sign}(Wq)$ and $h(x) = \text{sign}(Wx)$. **(2)** DP-RH+Hinge: We train random hyperplanes $W$ for these embeddings to compute the hashcodes as $g(q) = \text{sign}(W[0, q/||q||])$ and $h(x) = \text{sign}(W[\sqrt{T^2 - ||x||^2}, x])$. **(3)** WMH+Hinge: We use the best performing WMH implementation from DrHash toolkit [58] to obtain the hashcodes. **(4)** FLORA[15]: We train asymmetric hash networks (net$_1$, net$_2$) using an end-to-end data-driven approach, which minimizes bit balance and decorrelation loss, along with a consistency loss which predicts the final similarity score using $\cos(\text{net}_1(q), \text{net}_2(x))$.

Figure 3 compares the performance of FOURIERHASHNET, RH+Hinge, DP-RH+Hinge, WMH+Hinge and FLORA in terms of MAP for MSWEB and MSNBC datasets. Here we analyze the section of the trade-off curve which provides ≥10X speedup compared to exhaustive search. The complete tradeoff curve is provided in Appendix F. **(1)** The newly designed baselines are now seen to perform significantly better than those used in the previous experiments with Figure 2. However FOURIERHASHNET still outperforms all the baselines. **(2)** RH+Hinge, despite achieving the second highest scores in many cases, is seen to suffer from a large variance in performance within any given time budget. This would make it difficult to tune the hyperparameters to achieve the requisite performance v/s retrieval speed trade-off. **(3)** DP-RH+Hinge is seen to have a significantly worse performance than FOURIERHASHNET everywhere. This indicates that DP-RH is ill-suited to asymmetric hinge distance based retrieval.**(4)** We observe that for the same amount of query time invested, FLORA's MAP can lag ours by over 10%, particularly when faster average query times are required. FLORA's hyperparameter tuning is also more delicate, with there being unsuccessful settings (where MAP grows very slowly with query time) very close to relatively successful ones.

**Comparison with IVF indexing** We use the widely used FAISS-IVF [21] library, which supports IVF indexing based on L2 distance (IVF-L2) and Inner Product similarity (IVF-IP). Additionally, we propose an alternative Fourier+IVF+IP, where we apply Fourier transformation on the input embeddings, before using IVF-IP. We provide embeddings $q, x$, trained using hinge distance to all the methods, and use hinge distance to rank retrieved items.

Figure 4 compares the performances in terms of MAP, across all datasets. We observe that: **(1)** FOURIERHASHNET outperforms both IVF-L2 and IVF-IP across all datasets. FAISS-IVF retrieval suffers because its quantizers, that assign vectors to the Voronoi cells, rely on a metric like L2 or IP, which are unsuitable for asymmetric hinge distance. **(2)** Fourier transformation provides a significant boost in performance across all datasets, as seen while comparing Fourier+IVF+IP against IVF+IP. However, FOURIERHASHNET still outperforms Fourier+IVF+IP, most noticeably in MSWEB-2.

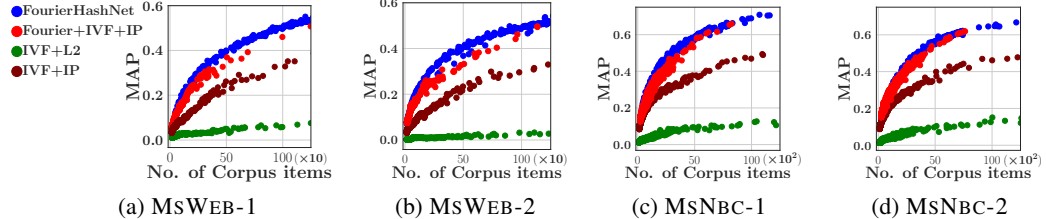

(a) MsWeb-1      (b) MsWeb-2      (c) MsNbc-1      (d) MsNbc-2

Figure 4: Trade-off between number of corpus items being evaluated and accuracy (MAP) for MsWeb and MsNbc datasets where there is $\geq$10X speedup compared to exhaustive search. We apply FourierHashNet and different IVF methods, *viz*, IVF+L2, IVF+IP, and Fourier+IVF+IP, on hinge distance guided embeddings; and then use the hinge distance to finally rank the retrieved items.

## 4.4 Ablation study

**Data driven vs data oblivious LSH** To perform ablation study on our proposed hashcode training method, we propose an alternative FHash (UN-TRAINED). This applies our Fourier features followed by a *data oblivious* random hyperplane LSH, without any data driven hashcode training.

In Figure 5, we compare the complete design of our method, *i.e.*, FourierHashNet and FHash (UNTRAINED) against the untrained versions of RH+Hinge and DP-RH+Hinge.

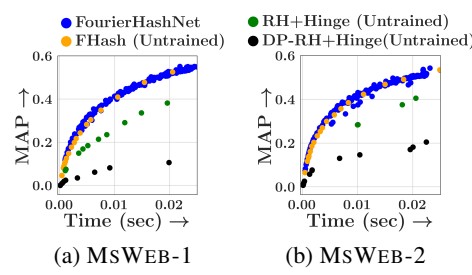

(a) MsWeb-1      (b) MsWeb-2

Figure 5: Effect of untrained RH

We make the following observations: **(1)** Benefit of Fourier transformation: The MAP vs time trade-off curve of FHash (UNTRAINED), consistently dominates all the baselines across both datasets. **(2)** Benefit of hashcode training: Compared to FHash (UNTRAINED), we observe that FourierHashNet allows for significantly more choices of trade-off points, where higher MAP is required.

**Ablation on collision minimizer** Here, we replace the collision minimizer in $\mathrm{loss}(Q, X \,|\, \boldsymbol{W})$ (14) with decorrelation loss which encourages hashcodes to be dissimilar: $\Delta_1 = \sum_{x \neq y} |\tanh(\boldsymbol{W}\boldsymbol{z}_x)^\top \tanh(\boldsymbol{W}\boldsymbol{z}_y)|$, a commonly used loss in prior work [54, 15].

Figure 6 compares the performance of the two variations of the losses in terms of MAP, for MsWeb datasets. We observe that: **(1)** Our loss containing the collision minimizer term performs better than its variant which uses the decorrelation loss. In MsWeb-2, latter provides a MAP of $0.4$ in $0.014$ secs, which our loss achieves in 50% of the time. **(2)** Our method allows for greater freedom in navigating the performance vs average query time trade-off, as seen in MsWeb-2, as it is more spread out across the time axis.

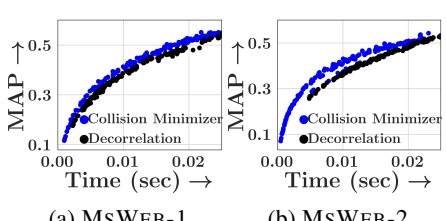

(a) MsWeb-1      (b) MsWeb-2

Figure 6: Collision minimizer vs. decorrelation.

## 5 Conclusion

We have presented FourierHashNet, an ALSH for asymmetric hinge distance, strongly motivated by text, image and graph retrieval applications. By converting hinge distance to a proposed dominance similarity and applying a suitable Fourier transform to the dominance similarity, we can estimate the distance as an inner product over an importance-sampled spectrum, which further enables the use of a trainable LSH in the frequency domain. Experiments show that FourierHashNet dramatically speeds up queries while preserving or improving retrieval accuracy. Our approach can be extended to any shift invariant functions including Chamfer distance, box embeddings, etc. Box embeddings are known to model more complex set operations like set overlap and set difference [42, 13], making them an interesting avenue for future research. One limitation of FourierHashNet compared to simple symmetric LSHs is the increase in computational cost to compute the Fourier transform. One can explore other types of transformations to mitigate this cost.

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

# Locality Sensitive Hashing in Fourier Frequency Domain For Soft Set Containment Search
## (Appendix)

## A    Limitations of our work

**(1)** In Eq. (9), the probability distribution $p(\boldsymbol{\omega})$ is determined based on the proportionality constant $||\boldsymbol{F}_\bullet(\iota\boldsymbol{\omega}^{1..M})||$, and we set $p(\boldsymbol{\omega}) = \prod_{k \in [K]} p(\omega)$, where $p(\omega) \propto |\text{Re}(S(\iota\omega))| + |\text{Im}(S(\iota\omega))|$. We note that this choice of distribution is not informed by the data distribution. Doing so may further improve FOURIERHASHNET.

**(2)** In our approach, the dominance similarity function is represented as an expectation of inner products of functions in the frequency domain, as described in Eq. (10). The accuracy of this representation relies on the quality of Monte Carlo approximations, which is influenced by the number of $\boldsymbol{\omega}$ samples used. Our experimental findings, presented in Figure13, suggest that it may be necessary to generate up to 100 samples per dimension to reduce the approximation error to acceptable levels. A better choice of $p(\boldsymbol{\omega})$ may reduce the number of samples needed.

**(3)** During our experimental investigations, we discovered that the computation of our proposed dominance similarity score is approximately 7.5X slower, compared to the dot product similarity on our datasets. This aligns with earlier observations where matrix subtraction operations have been known to be significantly slower than dot product. This disparity in computation speed represents another potential limitation of our approach, which could be addressed by exploring alternative designs for the dominance similarity function.

## B    Example applications of soft set containment

**Natural Language Inference (NLI)**    In (NLI) [4], $q$ and $x$ are sentences, regarded as sequences of words as items. A transformer network [14, 41] converts each sentence to an embedding vector. We infer $x \implies q$ if $\boldsymbol{q} \le \boldsymbol{x}$ [26]. Consider now a claim verification application that, given a claim as a query and a Web-scale corpus, needs to quickly retrieve passages that best support the given claim. This application exactly motivates FOURIERHASHNET.

**Market basket**    Given a basket of supermarket items, we may query purchase logs for frequent supersets to make recommendations. The corpus contains itemsets purchased in the past, each is a 'document' $x$. The query $q$ is the current basket. Hard set containment tests for $q \subset x$. However, we would like to 'soften' items in $q$ (say, from one toothpaste brand to another. Each item has a short textual description, which is passed into BERT [41] and the `[CLS]` embedding read out as the item representation. An itemset is then embedded using a (suitably fine-tuned) set encoder (such as Deep Set), giving us $\boldsymbol{q}$ and $\boldsymbol{x}$. To the extent $\boldsymbol{q} \le \boldsymbol{x}$, we regard the query basket as "soft-contained" in the document basket.

**Knowledge graph (KG) completion**    Vendrov et al. [52] embedded types and entities in a KG to vectors such that if two types are related via $t_1$ is-subtype-of $t_2$ then $\vec{t_1} \le \vec{t_2}$ was encouraged, and if entity $e$ is-instance-of $t$, then $\vec{e} \le \vec{t}$ was encouraged in suitable loss functions. Later, these *order* embeddings were generalized to *box* embeddings where is-subtype-of and is-instance-of were modeled as boxes in high dimensions contained in other boxes [10]. These models naturally motivate fast retrieval using dominance similarity.

**Subgraph isomorphism search**    Here we expect a corpus graph $x$ will be relevant if query graph $q$ is a *subgraph* of $x$, i.e., that $x$ has a subgraph that is *isomorphic* to $q$. In reality, we want to score highly corpus graphs that have subgraphs *almost* isomorphic to the query graph. A graph neural network (GNN) [25, 18] can build suitable contextual embeddings $\boldsymbol{q}$ and $\boldsymbol{x}$ for the entire graphs, which can be used to test for approximate subgraph isomorphism, *i.e.*, $\boldsymbol{q} \le \boldsymbol{x}$. There are several applications of subgraph isomorphism search. In material and drug design, there are large molecule databases. A researcher wishes to predict properties of a new query molecule by retrieving similar molecules in the database. Each molecule is modeled as a modest-sized graph with nodes (atom, DNA bases) and edges (valence, etc.). In image search [22], the query $q$ may be a graph fragment, e.g., ⟨person, feeding, pet⟩, and the goal is to find corpus graphs $x$ where $q$ is approximately a subgraph [31], *e.g.*, $x$ can contain ⟨man, feeding, dog⟩.

## C Further discussion on related work

In this section, we discuss existing work related to each of the three major components of our work—set embeddings, use of frequency domain for computing representations, and locality sensitive hashing.

**Neural set embeddings**    Motivated by various machine learning questions that are better formalized by using a set of items as primitive, there has also been a recent line work on set embeddings. Zaheer et al. [60] consider models that act on sets and characterize the structure of such permutation-invariant models, but do not consider asymmetric query based measures e.g., containment. Skianis et al. [48] casts the similarity measurement between sets as a combinatorial flow problem, which in turn is approximated by a linear program. Lee et al. [27] propose the Set Transformer, a model that uses self-attention to model interactions among the input set elements. Our model is different from the existing line of work in focusing on asymmetric metrics that can measure containment, as well as in using the frequency domain representation of the metric to build a scalable LSH.

**Application of frequency domain transformation in machine learning**    One of the most celebrated uses of the frequency domain representation was by [40], who proposed using random Fourier features for shift-invariant kernels. Since dot-product kernels are not shift-invariant, Bochner's theorem, a key tool in creating random Fourier features, does not apply. Hence, alternative random feature-based approximations have been proposed, primarily focusing on polynomial kernels [23, 38, 3, 49, 30]. All of the above work is on symmetric kernels. For our asymmetric dominance similarity function, we design a sampling distribution by taking into account the individual frequency-level coefficients of the Fourier representation. Raginsky and Lazebnik [39] provided a random Fourier based approach for LSH. However they only focused on similarity of the Mercer Kernel class, which is symmetric in nature.

**Locality sensitive hashing**    The third main pillar of our work is locality sensitive hashing (LSH) which enables efficient retrieval. Answering queries using sketches or hashes in order to measure the similarity or containment of documents has a long history, pioneered by Broder [6]. In more recent years, semantic search or vector search, sometimes called "dense passage retrieval" [24] employing scalable near-neighbor search engines, has emerged as a credible, often more powerful, alternative [34, 29] to standard information retrieval schemes, as the vector embeddings can capture more nuanced contextualization and semantics. While there are a number of variants of LSH, including multi-probe [32], our presentation and experimentation focus on a single-probe setting in which the hashing hyperplanes are learned from the data. In particular, we build upon the asymmetric hash constructions in Shrivastava and Li [46] and Neyshabur and Srebro [35].

# D Proofs of the technical results

## D.1 Proof of Proposition 3.1

*Proof.* Consider two functions $\text{BOX}_{a,b}$ (for arbitrary positive constants $a, b$) and $\text{RELU}_T$ defined as follows:

$$\text{BOX}_{a,b}(t) = \begin{cases} a \text{ if } -b \le t \le b \\ 0 \text{ else} \end{cases} \tag{16}$$

$$\text{RELU}_T(t) = \begin{cases} t \text{ if } 0 \le t \le T \\ 0 \text{ else} \end{cases} \tag{17}$$

Observe that $s$ can be written in terms of these new functions as follows: $s(t) = \text{BOX}_{T,T}(t) - \text{RELU}_T(t)$. By linearity of Fourier transform,

$$S(\iota\omega) = \mathcal{F}_{\text{BOX}_{T,T}}(\iota\omega) - \mathcal{F}_{\text{RELU}_T}(\iota\omega) \tag{18}$$

where, $\mathcal{F}_f(\iota\omega)$ denotes Fourier transform of $f(t)$ for any function $f$. Now let us compute $\mathcal{F}_{\text{RELU}_T}(\iota\omega)$.

$$\mathcal{F}_{\text{RELU}_T}(\iota\omega) = \frac{1}{2\pi} \int_0^T t e^{-\iota\omega t} dt \tag{19}$$

$$= -\frac{1}{2\pi\omega^2} + \frac{e^{-\iota\omega T}}{2\pi\omega^2} + \frac{\iota T e^{-\iota\omega T}}{2\pi\omega} \tag{20}$$

$$= -\frac{1}{2\pi\omega^2} + \frac{\cos(\omega T) - \iota\sin(\omega T)}{2\pi\omega^2} + \frac{T(\iota\cos(\omega T) + \sin(\omega T))}{2\pi\omega} \tag{21}$$

$$= \frac{-2\sin^2(\omega T/2)}{2\pi\omega^2} + \frac{T\sin(\omega T)}{2\pi\omega} - \iota\frac{\sin(\omega T)}{2\pi\omega^2} + \iota\frac{T\cos(\omega T)}{2\pi\omega} \tag{22}$$

Since $\text{BOX}_{T,T}(t)$ is a rectangular pulse, its Fourier transform is

$$\mathcal{F}_{\text{BOX}_{T,T}}(\iota\omega) = 2T\frac{\sin(\omega T)}{2\pi\omega} \tag{23}$$

Substituting the above Eqs. (22) and (23) into Eq. (18), we get $S(\iota\omega)$ as follows.

$$S(\iota\omega) = \underbrace{2T\frac{\sin(\omega T)}{2\pi\omega}}_{\text{BOX}_{T,T}(\iota\omega)} - \underbrace{\left[\frac{-2\sin^2(\omega T/2)}{2\pi\omega^2} + \frac{T\sin(\omega T)}{2\pi\omega} - \iota\frac{\sin(\omega T)}{2\pi\omega^2} + \iota\frac{T\cos(\omega T)}{2\pi\omega}\right]}_{G(\iota\omega)} \tag{24}$$

$$= \overbrace{T\frac{\sin(\omega T)}{2\pi\omega} + 2\frac{\sin^2(\frac{\omega T}{2})}{2\pi\omega^2}}^{\text{Re}(S(\iota\omega))} + \iota\underbrace{\left[\frac{\sin(\omega T)}{2\pi\omega^2} - \frac{T\cos(\omega T)}{2\pi\omega}\right]}_{\text{Im}(S(\iota\omega))} \tag{25}$$

$\square$

## D.2 Proof of Theorem 3.2

**Theorem 3.2.** Let $q, x \in \mathbb{R}^K$, $\cos^{-1}$ be Lipschitz with Lipschitz constant $L_{\cos}$; the hyperparameter $T$ in Eq. (4) be chosen such that $T > ||q - x||_\infty$; the frequency sampling distribution $p(\omega_k^j) \propto [|\text{Re}(S(\omega_k^j))| + |\text{Im}(S(\omega_k^j))|]$ with the support set $\omega_k^j \in [-\omega_{\max}, \omega_{\max}]$. Furthermore, we define $\Delta_{\text{sim}} := \frac{T}{T - \max_{x,q} ||x - q||_\infty}$ and $I(\omega_{\max}) = \int_{-\omega_{\max}}^{\omega_{\max}} [|\text{Re}(S(\omega)| + |\text{Im}(S(\omega))|] d\omega$. Then, the mapping $g(q)[i] = \text{sign}(\mathbf{w}_i^\top \mathbf{F}_q(\boldsymbol{\omega}^{1...M}))$ and $h(x)[i] = \text{sign}(\mathbf{w}_i^\top \mathbf{F}_x(\boldsymbol{\omega}^{1...M}))$ where $\mathbf{w}_i \sim N(0, \mathbb{I})$, constitutes a $(S_0, cS_0, p_1, p_2)$-ALSH for some $p_1$ and $p_2$ if

$$\omega_{\max} > \frac{4\Delta_{\text{sim}}}{\pi(1-c)S_0}; \qquad M > \left[\frac{4L_{\cos}}{(1-c)S_0}\right]^2 KI(\omega_{\max}). \tag{26}$$

**Proof** Define $\Omega = [-\omega_{\max}, \omega_{\max}]^K$. For $\boldsymbol{\omega} \in \Omega$, we have the following relationships:

$$||\mathbf{F}_q(\boldsymbol{\omega}^{1...M})||_2^2 = ||\mathbf{F}_x(\boldsymbol{\omega}^{1...M})||_2^2 = \sum_{i \in [M], k \in [K]} \frac{|Re(S(\omega_k^j))| + |Im(S(\omega_k^j))|}{p(\omega_k^j)} = MKI; \tag{27}$$

We also define that $\widehat{\text{sim}}_{\omega_{\max}}(q, x) = \int_{\boldsymbol{\omega} \in \Omega} \boldsymbol{F}_q(\iota\boldsymbol{\omega})^\top \boldsymbol{F}_x(\iota\boldsymbol{\omega}) p(\boldsymbol{\omega}) d\boldsymbol{\omega}$. This gives us the following result:

$$\Pr_{g,h}[g(q) = h(x) | \boldsymbol{\omega}^i] = 1 - \frac{1}{\pi} \cos^{-1} \left( \frac{\boldsymbol{F}_q(\boldsymbol{\omega}^{1...M})^\top \boldsymbol{F}_x(\boldsymbol{\omega}^{1...M})}{||\boldsymbol{F}_q(\boldsymbol{\omega}^{1...M})|| ||\boldsymbol{F}_x(\boldsymbol{\omega}^{1...M})||} \right) \tag{28}$$

$$= 1 - \frac{1}{\pi} \cos^{-1} \left( \int_{\boldsymbol{\omega} \in \Omega} \frac{\boldsymbol{F}_q(\iota\boldsymbol{\omega})^\top \boldsymbol{F}_x(\iota\boldsymbol{\omega}) p(\boldsymbol{\omega}) d\boldsymbol{\omega}}{||\boldsymbol{F}_q(\boldsymbol{\omega})|| ||\boldsymbol{F}_x(\boldsymbol{\omega})||} \right)$$
$$- \frac{1}{\pi} \cos^{-1} \left( \frac{\boldsymbol{F}_q(\boldsymbol{\omega}^{1...M})^\top \boldsymbol{F}_x(\boldsymbol{\omega}^{1...M})}{MKI(\omega_{\max})} \right) + \frac{1}{\pi} \cos^{-1} \left( \int_{\boldsymbol{\omega} \in \Omega} \frac{\boldsymbol{F}_q(\iota\boldsymbol{\omega})^\top \boldsymbol{F}_x(\iota\boldsymbol{\omega}) p(\boldsymbol{\omega}) d\boldsymbol{\omega}}{\underbrace{||\boldsymbol{F}_q(\boldsymbol{\omega})|| ||\boldsymbol{F}_x(\boldsymbol{\omega})||}_{KI(\omega_{\max})}} \right) \tag{29}$$

$$= 1 - \frac{1}{\pi} \cos^{-1} \left( \frac{\text{sim}(q, x)}{KI(\omega_{\max})} \right) + \underbrace{\frac{1}{\pi} \cos^{-1} \left( \frac{\text{sim}(q, x)}{KI(\omega_{\max})} \right) - \frac{1}{\pi} \cos^{-1} \left( \frac{\widehat{\text{sim}}_{\omega_{\max}}(q, x)}{KI(\omega_{\max})} \right)}_{\text{Term-1}}$$
$$\underbrace{- \frac{1}{\pi} \cos^{-1} \left( \frac{\boldsymbol{F}_q(\boldsymbol{\omega}^{1...M})^\top \boldsymbol{F}_x(\boldsymbol{\omega}^{1...M})}{MKI(\omega_{\max})} \right) + \frac{1}{\pi} \cos^{-1} \left( \int_{\boldsymbol{\omega} \in \Omega} \frac{\boldsymbol{F}_q(\iota\boldsymbol{\omega})^\top \boldsymbol{F}_x(\iota\boldsymbol{\omega}) p(\boldsymbol{\omega}) d\boldsymbol{\omega}}{KI(\omega_{\max})} \right)}_{\text{Term-2}} \tag{30}$$

Next we bound Term-1 and Term-2. First we bound Term-1 as follows:

$$\left| \frac{1}{\pi} \left[ \cos^{-1} \left( \frac{\text{sim}(q, x)}{KI(\omega_{\max})} \right) - \cos^{-1} \left( \frac{\widehat{\text{sim}}_{\omega_{\max}}(q, x)}{KI(\omega_{\max})} \right) \right] \right| \le \frac{L_{\cos}}{\pi KI} \left| \text{sim}(q, x) - \widehat{\text{sim}}_{\omega_{\max}}(q, x) \right| \tag{31}$$

$$\le \frac{KL_{\cos} \underbrace{\left( 6 + \frac{2T}{T - \max_{\boldsymbol{x}, \boldsymbol{q}} ||\boldsymbol{x} - \boldsymbol{q}||_\infty} \right)}_{\Delta_{\text{sim}}}}{\pi^2 KI(\omega_{\max}) \omega_{\max}} \tag{32}$$

The last inequality is due to Lemma D.1. Next we bound Term-2 in Eq. (30).

$$\frac{1}{\pi} \mathbb{E} \left[ \left| -\cos^{-1} \left( \frac{\boldsymbol{F}_q(\boldsymbol{\omega}^{1...M})^\top \boldsymbol{F}_x(\boldsymbol{\omega}^{1...M})}{MKI(\omega_{\max})} \right) + \cos^{-1} \left( \int_{\boldsymbol{\omega} \in \Omega} \frac{\boldsymbol{F}_q(\iota\boldsymbol{\omega})^\top \boldsymbol{F}_x(\iota\boldsymbol{\omega}) p(\boldsymbol{\omega}) d\boldsymbol{\omega}}{KI(\omega_{\max})} \right) \right| \right] \tag{33}$$

$$\le \frac{L_{\cos}}{\pi KI} \left| \boldsymbol{F}_q(\boldsymbol{\omega}^{1...M})^\top \boldsymbol{F}_x(\boldsymbol{\omega}^{1...M}) / M - \int_{\boldsymbol{\omega} \in \Omega} \boldsymbol{F}_q(\iota\boldsymbol{\omega})^\top \boldsymbol{F}_x(\iota\boldsymbol{\omega}) p(\boldsymbol{\omega}) d\boldsymbol{\omega} \right| \tag{34}$$

$$= \frac{L_{\cos}}{\pi KI(\omega_{\max})} \left| \sum_{j \in [M]} \boldsymbol{F}_q(\boldsymbol{\omega}^j)^\top \boldsymbol{F}_x(\boldsymbol{\omega}^j) / M - \int_{\boldsymbol{\omega} \in \Omega} \boldsymbol{F}_q(\iota\boldsymbol{\omega})^\top \boldsymbol{F}_x(\iota\boldsymbol{\omega}) p(\boldsymbol{\omega}) d\boldsymbol{\omega} \right| \tag{35}$$

Taking expectation with respect to $\boldsymbol{\omega}$, we have:

$$\frac{1}{\pi} \mathbb{E} \left[ \left| -\cos^{-1} \left( \frac{\boldsymbol{F}_q(\boldsymbol{\omega}^{1...M})^\top \boldsymbol{F}_x(\boldsymbol{\omega}^{1...M})}{MKI(\omega_{\max})} \right) + \cos^{-1} \left( \int_\omega \frac{\boldsymbol{F}_q(\iota\boldsymbol{\omega})^\top \boldsymbol{F}_x(\iota\boldsymbol{\omega}) p(\boldsymbol{\omega}) d\boldsymbol{\omega}}{KI(\omega_{\max})} \right) \right| \right] \tag{36}$$

$$\le \frac{L_{\cos}}{\pi KI(\omega_{\max})} \mathbb{E} \left| \sum_{j \in [M]} \boldsymbol{F}_q(\boldsymbol{\omega}^j)^\top \boldsymbol{F}_x(\boldsymbol{\omega}^j) / M - \int_\omega \boldsymbol{F}_q(\iota\boldsymbol{\omega})^\top \boldsymbol{F}_x(\iota\boldsymbol{\omega}) p(\boldsymbol{\omega}) d\boldsymbol{\omega} \right| \tag{37}$$

$$\le \frac{L_{\cos}}{\pi KI(\omega_{\max})} \left( \text{Variance} \left[ \sum_{j \in [M]} \boldsymbol{F}_q(\boldsymbol{\omega}^j)^\top \boldsymbol{F}_x(\boldsymbol{\omega}^j) / M \right] \right)^{1/2} \quad (\mathbb{E}[|Z|] \le \sqrt{\mathbb{E}[|Z|^2]}) \tag{38}$$

$$= \frac{L_{\cos}}{\pi KI(\omega_{\max}) \sqrt{M}} \left( \text{Variance} \left[ \boldsymbol{F}_q(\iota\boldsymbol{\omega})^\top \boldsymbol{F}_x(\iota\boldsymbol{\omega}) \right] \right)^{1/2} \tag{39}$$

$$= \frac{L_{\cos}}{\pi KI(\omega_{\max}) \sqrt{M}} \sqrt{K} \left( \text{Variance} \left[ \boldsymbol{F}_q(\omega_k)^\top \boldsymbol{F}_x(\omega_k) \right] \right)^{1/2} \le \frac{L_{\cos}}{\pi \sqrt{KM}} \tag{40}$$

The last inequality follows from bound on the variance due the following:

$$\boldsymbol{F}_q(\omega_k)^\top \boldsymbol{F}_x(\omega_k) = \boldsymbol{S}_q(\omega_k)^\top \boldsymbol{S}_x(\omega_k)/p(\omega_k) \tag{41}$$

$$= \frac{Re[\boldsymbol{S}(\omega)]\cos\omega_k(q[k]-x[k]) - Im[\boldsymbol{S}(\omega)]\sin\omega_k(q[k]-x[k]))}{(1/I)[|Re[\boldsymbol{S}(\omega)]| + |Im[\boldsymbol{S}(\omega)]|]} \tag{42}$$

$$\leq I\frac{|Re[\boldsymbol{S}(\omega_k)]| + |Im[\boldsymbol{S}(\omega_k)]|}{|Re[\boldsymbol{S}(\omega)]| + |Im[\boldsymbol{S}(\omega)]|} = I \tag{43}$$

Putting Eqs. (32) (40) into Eq. (30), we will have the upper bound for $\text{sim}(q,x) \leq cS_0$:

$$\mathbb{E}[\Pr_{g,h}[g(q) = h(x)|\boldsymbol{\omega}^j]] \leq p_2 = 1 - \frac{1}{\pi}\cos^{-1}\left(\frac{\text{sim}(q,x)}{KI(\omega_{\max})}\right) + \frac{L_{\cos}}{\pi\sqrt{KM}} + \frac{\Delta_{\text{sim}}}{\pi^2 KI(\omega_{\max})\omega_{\max}} \tag{44}$$

Similarly, we have the lower bound for $\text{sim}(q,x) > S_0$:

$$\mathbb{E}[\Pr_{g,h}[g(q) = h(x)|\boldsymbol{\omega}^j]] \geq p_1 = 1 - \frac{1}{\pi}\cos^{-1}\left(\frac{\text{sim}(q,x)}{KI(\omega_{\max})}\right) - \frac{L_{\cos}}{\pi\sqrt{KM}} - \frac{\Delta_{\text{sim}}}{\pi^2 KI(\omega_{\max})\omega_{\max}} \tag{45}$$

Now, note that:

$$\frac{1}{\pi}\left[\cos^{-1}\left(\frac{cS_0}{KI(\omega_{\max})}\right) - \cos^{-1}\left(\frac{S_0}{KI(\omega_{\max})}\right)\right] \geq \frac{1}{\pi KI(\omega_{\max})}(c-1)S_0 \left.\frac{d}{ds}\cos^{-1}(s)\right|_{s\in(cS_0,S_0)}$$

$$= \frac{1}{\pi KI(\omega_{\max})}(c-1)S_0 \times -\frac{1}{\sqrt{1-s^2}}$$

$$= \frac{(1-c)S_0}{\pi KI(\omega_{\max})\sqrt{1-s^2}}$$

$$\geq \frac{(1-c)S_0}{\pi KI(\omega_{\max})}$$

$$\overset{(i)}{\geq} \frac{2\Delta_{\text{sim}}}{\pi^2 KI(\omega_{\max})\omega_{\max}} + \frac{2L_{\cos}}{\pi\sqrt{KI(\omega_{\max})M}} \tag{46}$$

The last inequality (i) is due to the conditions that:

$$\omega_{\max} > \frac{4\Delta_{\text{sim}}}{\pi(1-c)S_0}; \qquad M > \left[\frac{4L_{\cos}}{(1-c)S_0}\right]^2 KI(\omega_{\max}). \tag{47}$$

This leads us to the following relationship:

$$1 - \frac{1}{\pi}\cos^{-1}\left(\frac{cS_0}{KI(\omega_{\max})}\right) + \frac{L_{\cos}}{\pi\sqrt{KM}} + \frac{\Delta_{\text{sim}}}{\pi^2 KI(\omega_{\max})\omega_{\max}}$$

$$\leq 1 - \frac{1}{\pi}\cos^{-1}\left(\frac{S_0}{KI(\omega_{\max})}\right) - \frac{L_{\cos}}{\pi\sqrt{KM}} - \frac{\Delta_{\text{sim}}}{\pi^2 KI(\omega_{\max})\omega_{\max}} \tag{48}$$

Thus, we have:

$$p_2 = 1 - \frac{1}{\pi}\cos^{-1}\left(\left.\frac{\text{sim}(q,x)}{KI(\omega_{\max})}\right|_{\text{sim}(q,x)<cS_0}\right) + \frac{L_{\cos}}{\pi\sqrt{KM}} + \frac{\Delta_{\text{sim}}}{\pi^2 KI(\omega_{\max})\omega_{\max}}$$

$$\leq 1 - \frac{1}{\pi}\cos^{-1}\left(\frac{cS_0}{KI(\omega_{\max})}\right) + \frac{L_{\cos}}{\pi\sqrt{KM}} + \frac{\Delta_{\text{sim}}}{\pi^2 KI(\omega_{\max})\omega_{\max}}$$

$$\leq 1 - \frac{1}{\pi}\cos^{-1}\left(\frac{S_0}{KI(\omega_{\max})}\right) - \frac{L_{\cos}}{\pi\sqrt{KM}} - \frac{\Delta_{\text{sim}}}{\pi^2 KI(\omega_{\max})\omega_{\max}}$$

$$\leq 1 - \frac{1}{\pi}\cos^{-1}\left(\left.\frac{\text{sim}(q,x)}{KI(\omega_{\max})}\right|_{\text{sim}(q,x)>S_0}\right) - \frac{L_{\cos}}{\pi\sqrt{KM}} - \frac{\Delta_{\text{sim}}}{\pi^2 KI(\omega_{\max})\omega_{\max}}$$

$$= p_1 \tag{49}$$

### D.3 Effect of truncating frequencies

**Lemma D.1.** Let $\widehat{\text{sim}}_{\omega_{\max}}(q, x) = \mathbb{E}_{\boldsymbol{\omega} \sim p(\boldsymbol{\omega})}[\boldsymbol{F}_q(\iota\boldsymbol{\omega})^\top \boldsymbol{F}_x(\iota\boldsymbol{\omega})]$ with $p(\boldsymbol{\omega})$ being defined as follows:

$$p(\boldsymbol{\omega}) \propto \begin{cases} \prod_{k \in [K]} |\text{Re}(S(\iota\omega_k))| + |\text{Im}(S(\iota\omega_k))| & \text{if, } \omega \in [-\omega_{\max}, \omega_{\max}] \\ 0 & \text{otherwise} \end{cases} \tag{50}$$

Then, if $T > \max_{\boldsymbol{x}, \boldsymbol{q}} ||\boldsymbol{x} - \boldsymbol{q}||_\infty$, then we have:

$$|\widehat{\text{sim}}_{\omega_{\max}}(q, x) - \text{sim}(q, x)| \leq \frac{6K + 2K \max\left\{1, \dfrac{T}{T - \max_{\boldsymbol{x}, \boldsymbol{q}} ||\boldsymbol{x} - \boldsymbol{q}||_\infty}\right\}}{\pi \omega_{\max}} \tag{51}$$

**Proof**   Let $S(\iota\omega)$ be the Fourier transform of $s(t)$, $\mathcal{F}^{-1}$ is the inverse Fourier transform, $\star$ is the convolution operator. Further, we define the $\text{LOWPASS}(\omega \mid \omega_{\max})$ is defined as

$$\text{LOWPASS}(\omega \mid \omega_{\max}) = 1 \text{ if } |\omega| \leq \omega_{\max} \text{ and } 0 \text{ otherwise.} \tag{52}$$

Let us define the filtered or frequency truncated signal of $s(t)$ as defined in Eq. (4) as:

$$\widehat{s}_{\omega_{\max}}(t) = \mathcal{F}^{-1}\Big[S(\iota\omega)\text{LOWPASS}(\omega \mid \omega_{\max})\Big]$$
$$= \mathcal{F}^{-1}[S(\iota\omega)] \star \mathcal{F}^{-1}[\text{LOWPASS}(\omega \mid \omega_{\max})]. \tag{53}$$

In this context, $\widehat{\text{sim}}_{\omega_{\max}}(q, x)$ can be expressed as follows:

$$\widehat{\text{sim}}_{\omega_{\max}}(q, x)$$

$$= \int_{\boldsymbol{\omega} \in \Omega} \boldsymbol{F}_q(\iota\boldsymbol{\omega})^\top \boldsymbol{F}_x(\iota\boldsymbol{\omega}) p(\boldsymbol{\omega}) d\boldsymbol{\omega} \tag{54}$$

$$= \int_{\boldsymbol{\omega} \in \Omega} \boldsymbol{S}_q(\iota\boldsymbol{\omega})^\top \boldsymbol{S}_c(\iota\boldsymbol{\omega}) d\boldsymbol{\omega} \tag{55}$$

$$= \sum_{k \in [K]} \int_{-\omega_{\max}}^{\omega_{\max}} [\text{Re}(S(\iota\omega_k)) \cos(\omega_k(\boldsymbol{q}[k] - \boldsymbol{x}[k])) - \text{Im}(S(\iota\omega_k)) \sin(\omega_k(\boldsymbol{q}[k] - \boldsymbol{x}[k]))] d\omega_k \tag{56}$$

$$\stackrel{(i)}{=} \sum_{k \in [K]} \int_{-\omega_{\max}}^{\omega_{\max}} [\text{Re}(S(\iota\omega_k)) \cos(\omega_k(\boldsymbol{q}[k] - \boldsymbol{x}[k])) - \text{Im}(S(\iota\omega_k)) \sin(\omega_k(\boldsymbol{q}[k] - \boldsymbol{x}[k]))] d\omega_k$$
$$+ \sum_{k \in [K]} \iota \int_{-\omega_{\max}}^{\omega_{\max}} \underbrace{[\text{Im}(S(\iota\omega_k)) \cos(\omega_k(\boldsymbol{q}[k] - \boldsymbol{x}[k])) + \text{Re}(S(\iota\omega_k)) \sin(\omega_k(\boldsymbol{q}[k] - \boldsymbol{x}[k]))]}_{\text{Odd function in } \omega_k} d\omega_k \tag{57}$$

$$= \sum_{k \in [K]} \int_{-\omega_{\max}}^{\omega_{\max}} [\text{Re}(S(\iota\omega_k)) + \iota\text{Im}(S(\iota\omega_k)) \exp(\iota\omega_k(\boldsymbol{q}[k] - \boldsymbol{x}[k])) d\omega_k \tag{58}$$

$$= \sum_{k \in [K]} \int_{-\omega_{\max}}^{\omega_{\max}} S(\iota\omega_k) \exp(\iota\omega_k(\boldsymbol{q}[k] - \boldsymbol{x}[k])) d\omega_k \tag{59}$$

$$= \sum_{k \in [K]} \int_{-\infty}^{\infty} S(\iota\omega_k) \text{LOWPASS}(\omega_k \mid \omega_{\max}) \exp(\omega_k(\boldsymbol{q}[k] - \boldsymbol{x}[k])) d\omega_k \tag{60}$$

$$= \sum_{k \in [K]} \mathcal{F}^{-1}[S(\iota\omega_k) \text{LOWPASS}(\omega_k \mid \omega_{\max})] \tag{61}$$

$$\stackrel{(ii)}{=} \sum_{k \in [K]} \widehat{s}_{\omega_{\max}}(\boldsymbol{q}[k] - \boldsymbol{x}[k]) \tag{62}$$

Equality (i) is due to the fact that the newly added imaginary integrand is an odd function (since, $\text{Im}(S(\iota\omega)) = -\text{Im}(S(-\iota\omega))$ and $\text{Re}(S(\iota\omega)) = \text{Re}(S(-\iota\omega))$) and thus, the integral from $-\omega_{\max}$ and $\omega_{\max}$ is zero. Equality (ii) is due to Eq. (53). Next we focus on $\widehat{s}_{\omega_{\max}}(t)$. Since $\mathcal{F}^{-1}[S(\iota\omega)] =$

$s(t)$ and $\mathcal{F}^{-1}[\text{LowPass}(\omega \mid \omega_{\max}) = \frac{\sin \omega_{\max} t}{\pi t}$, we have:

$$\widehat{s}_{\omega_{\max}}(t) = \int_{-\infty}^{\infty} s(t-\tau) \frac{\sin \omega_{\max} \tau}{\pi \tau} d\tau \tag{63}$$

$$= \int_{-\infty}^{\infty} s\left(t - \frac{\tau}{\omega_{\max}}\right) \frac{\sin \tau}{\pi \tau} d\tau \tag{64}$$

$$= \int_{-\infty}^{\infty} s(t) \frac{\sin \tau}{\pi \tau} d\tau + \underbrace{\left[s\left(t - \frac{\tau}{\omega_{\max}}\right) - s(t)\right]}_{\Delta s\left(t - \frac{\tau}{\omega_{\max}}\right)} \frac{\sin \tau}{\pi \tau} d\tau \tag{65}$$

$$= s(t) \int_{-\infty}^{\infty} \frac{\sin \tau}{\pi \tau} d\tau + \int_{-\infty}^{\infty} \underbrace{\left[s\left(t - \frac{\tau}{\omega_{\max}}\right) - s(t)\right]}_{\Delta s\left(t - \frac{\tau}{\omega_{\max}}\right)} \frac{\sin \tau}{\pi \tau} d\tau \tag{66}$$

$$= s(t) + \int_{-\infty}^{\infty} \left[s\left(t - \frac{\tau}{\omega_{\max}}\right) - s(t)\right] \frac{\sin \tau}{\pi \tau} d\tau \tag{67}$$

We only consider the case when $t \in (-T, T)$. We decompose the error term $\widehat{s}_{\omega_{\max}}(t) - s(t)$ as:

$$\widehat{s}_{\omega_{\max}}(t) - s(t) = \int_{-\infty}^{\omega_{\max}(t-T)} \Delta s\left(t - \frac{\tau}{\omega_{\max}}\right) \frac{\sin \tau}{\pi \tau} d\tau + \int_{\omega_{\max}(t-T)}^{\omega_{\max} t} \Delta s\left(t - \frac{\tau}{\omega_{\max}}\right) \frac{\sin \tau}{\pi \tau} d\tau \tag{68}$$

$$+ \int_{\omega_{\max} t}^{\omega_{\max}(t+T)} \Delta s\left(t - \frac{\tau}{\omega_{\max}}\right) \frac{\sin \tau}{\pi \tau} d\tau + \int_{\omega_{\max}(t+T)}^{\infty} \Delta s\left(t - \frac{\tau}{\omega_{\max}}\right) \frac{\sin \tau}{\pi \tau} d\tau \tag{69}$$

**Bounding the error** $|\widehat{s}_{\omega_{\max}}(t) - s(t)|$ **when** $-T < t \le 0$   We separately bound the four integrals as follows:

*Bound on the first integral* $\int_{-\infty}^{\omega_{\max}(t-T)} \Delta s\left(t - \frac{\tau}{\omega_{\max}}\right) \frac{\sin \tau}{\pi \tau} d\tau$: Thus, $s\left(t - \frac{\tau}{\omega_{\max}}\right) = 0$ and $\Delta s\left(t - \frac{\tau}{\omega_{\max}}\right) = s\left(t - \frac{\tau}{\omega_{\max}}\right) - s(t) = 0 - T$. We compute the bound on the first term as:

$$\int_{-\infty}^{\omega_{\max}(t-T)} \Delta s\left(t - \frac{\tau}{\omega_{\max}}\right) \frac{\sin \tau}{\pi \tau} d\tau = -T \int_{-\infty}^{\omega_{\max}(t-T)} \frac{\sin \tau}{\pi \tau} d\tau \tag{70}$$

$$= -T \int_{\omega_{\max}(T-t)}^{\infty} \frac{\sin \tau}{\pi \tau} d\tau \tag{71}$$

Thus, from Proposition D.2, we have:

$$\left| \int_{-\infty}^{\omega_{\max}(t-T)} \Delta s\left(t - \frac{\tau}{\omega_{\max}}\right) \frac{\sin \tau}{\pi \tau} d\tau \right| \le \frac{2T}{\pi \omega_{\max}(T - t)} \le \frac{2}{\pi \omega_{\max}} \quad (\text{since } t < 0) \tag{72}$$

*Bound on the second integral* $\int_{\omega_{\max}(t-T)}^{\omega_{\max} t} \Delta s\left(t - \frac{\tau}{\omega_{\max}}\right) \frac{\sin \tau}{\pi \tau} d\tau$: Since $\tau \in (\omega_{\max}(t-T), \omega_{\max} t]$, we have $0 \le t - \tau/\omega_{\max} < T$. Thus, $s\left(t - \frac{\tau}{\omega_{\max}}\right) = T - t + \frac{\tau}{\omega_{\max}}$.

We compute the bound on the second term as:

$$\int_{\omega_{\max}(t-T)}^{\omega_{\max} t} \Delta s\left(t - \frac{\tau}{\omega_{\max}}\right) \frac{\sin \tau}{\pi \tau} d\tau \tag{73}$$

$$= \int_{\omega_{\max}(t-T)}^{\omega_{\max} t} \left(T - t + \frac{\tau}{\omega_{\max}} - T\right) \frac{\sin \tau}{\pi \tau} d\tau \tag{74}$$

$$= -t \int_{\omega_{\max}(t-T)}^{\omega_{\max} t} \frac{\sin \tau}{\pi \tau} d\tau + \int_{\omega_{\max}(t-T)}^{\omega_{\max} t} \frac{\tau}{\omega_{\max}} \frac{\sin \tau}{\pi \tau} d\tau \tag{75}$$

$$= -t \int_{\omega_{\max}(t-T)}^{\omega_{\max} t} \frac{\sin \tau}{\pi \tau} d\tau + \frac{\cos \omega_{\max}(t-T) - \cos \omega_{\max} t}{\pi \omega_{\max}} \tag{76}$$

Thus, from Proposition D.2, we have:

$$\left| \int_{\omega_{\max}(t-T)}^{\omega_{\max}t} \Delta s\left(t - \frac{\tau}{\omega_{\max}}\right) \frac{\sin \tau}{\pi \tau} d\tau \right| \leq \frac{2t}{\pi \omega_{\max} t} + \frac{2}{\pi \omega_{\max}} = \frac{4}{\pi \omega_{\max}} \tag{77}$$

*Bound on the third integral* $\int_{\omega_{\max}t}^{\omega_{\max}(t+T)} \Delta s\left(t - \frac{\tau}{\omega_{\max}}\right) \frac{\sin \tau}{\pi \tau} d\tau$: When $\tau \in (\omega_{\max}t, \omega_{\max}(t+T)]$, we have $-T \leq t - \tau/\omega_{\max} < 0$. Moreover, $-T \leq t < 0$. Thus, $s(t) = s(t - \tau/\omega_{\max}) = T$, resulting in $\Delta s\left(t - \frac{\tau}{\omega_{\max}}\right) = 0$ and the third integral is zero.

$$\int_{\omega_{\max}t}^{\omega_{\max}(t+T)} \Delta s\left(t - \frac{\tau}{\omega_{\max}}\right) \frac{\sin \tau}{\pi \tau} d\tau = 0 \tag{78}$$

*Bound on the fourth integral* $\int_{\omega_{\max}(t+T)}^{\infty} \Delta s\left(t - \frac{\tau}{\omega_{\max}}\right) \frac{\sin \tau}{\pi \tau} d\tau$: When $\tau \in (\omega_{\max}(t + T), \infty)$, we have $-\infty < t - \tau/\omega_{\max} < -T$. Then, $s\left(t - \frac{\tau}{\omega_{\max}}\right) = 0$.

We compute the bound on the first term as:

$$\int_{\omega_{\max}(T+t)}^{\infty} \Delta s\left(t - \frac{\tau}{\omega_{\max}}\right) \frac{\sin \tau}{\pi \tau} d\tau = -T \int_{\omega_{\max}(T+t)}^{\infty} \frac{\sin \tau}{\pi \tau} d\tau \tag{79}$$

$$\tag{80}$$

Thus, from Proposition D.2, we have:

$$\left| \int_{\omega_{\max}(T+t)}^{\infty} \Delta s\left(t - \frac{\tau}{\omega_{\max}}\right) \frac{\sin \tau}{\pi \tau} d\tau \right| \leq \frac{2T}{\pi \omega_{\max}(T + t)} \tag{81}$$

**Bounding the error** $|\widehat{s}_{\omega_{\max}}(t) - s(t)|$ **when** $0 \leq t < T$

$$\widehat{s}_{\omega_{\max}}(t) - s(t) = \int_{-\infty}^{\omega_{\max}(t-T)} \Delta s\left(t - \frac{\tau}{\omega_{\max}}\right) \frac{\sin \tau}{\pi \tau} d\tau + \int_{\omega_{\max}(t-T)}^{\omega_{\max}t} \Delta s\left(t - \frac{\tau}{\omega_{\max}}\right) \frac{\sin \tau}{\pi \tau} d\tau \tag{82}$$

$$+ \int_{\omega_{\max}t}^{\infty} \Delta s\left(t - \frac{\tau}{\omega_{\max}}\right) \frac{\sin \tau}{\pi \tau} d\tau \tag{83}$$

Now, since $0 \leq t < T$, we have $s(t) = T - t$, throughout the next part of the proof.

*Bound on the first integral* $\int_{-\infty}^{\omega_{\max}(t-T)} \Delta s\left(t - \frac{\tau}{\omega_{\max}}\right) \frac{\sin \tau}{\pi \tau} d\tau$: Since $\tau \in (-\infty, \omega_{\max}(t - T)]$, we have $T \leq t - \tau/\omega_{\max} < \infty$. Thus, $s\left(t - \frac{\tau}{\omega_{\max}}\right) = 0$ and $\Delta s\left(t - \frac{\tau}{\omega_{\max}}\right) = s\left(t - \frac{\tau}{\omega_{\max}}\right) - s(t) = 0 - (T - t)$.

$$\left| \int_{-\infty}^{\omega_{\max}(t-T)} \Delta s\left(t - \frac{\tau}{\omega_{\max}}\right) \frac{\sin \tau}{\pi \tau} d\tau \right|$$

$$= \left| -(T - t) \int_{-\infty}^{\omega_{\max}(t-T)} \frac{\sin \tau}{\pi \tau} d\tau \right| \overset{(i)}{\leq} (T - t) \frac{2}{\omega_{\max}(T - t)\pi} = \frac{2}{\omega_{\max}\pi} \tag{84}$$

Inequality (i) is due to Proposition D.2

*Bound on the second integral* $\int_{\omega_{\max}(t-T)}^{\omega_{\max}t} \Delta s\left(t - \frac{\tau}{\omega_{\max}}\right) \frac{\sin \tau}{\pi \tau} d\tau$: Since $\tau \in (\omega_{\max}(t - T), \omega_{\max}t]$, we have $0 \leq t - \tau/\omega_{\max} < T$. Thus, $s\left(t - \frac{\tau}{\omega_{\max}}\right) = T - t + \frac{\tau}{\omega_{\max}}$.

$$\left| \int_{\omega_{\max}(t-T)}^{\omega_{\max}t} \Delta s\left(t - \frac{\tau}{\omega_{\max}}\right) \frac{\sin \tau}{\pi \tau} d\tau \right| = \left| \int_{\omega_{\max}(t-T)}^{\omega_{\max}t} [T - (t - \tau/\omega_{\max}) - (T - t)] \frac{\sin \tau}{\pi \tau} \right|$$

$$= \left| \int_{\omega_{\max}(t-T)}^{\omega_{\max}t} \frac{\sin \tau}{\pi \omega_{\max}} d\tau \right| \leq \frac{2}{\pi \omega_{\max}} \tag{85}$$

*Bound on the third integral* $\int_{\omega_{\max}t}^{\omega_{\max}(t+T)} \Delta s\left(t - \frac{\tau}{\omega_{\max}}\right) \frac{\sin\tau}{\pi\tau} d\tau$*: When* $\tau \in (\omega_{\max}t, \omega_{\max}(t+T)]$, we have $-T \leq t - \tau/\omega_{\max} < 0$. Thus, $s(t - \tau/\omega_{\max}) = T$.

$$\left|\int_{\omega_{\max}t}^{\omega_{\max}(t+T)} \Delta s\left(t - \frac{\tau}{\omega_{\max}}\right) \frac{\sin\tau}{\pi\tau} d\tau\right| = \left|\int_{\omega_{\max}t}^{\omega_{\max}(t+T)} [T - (T-t)]\frac{\sin\tau}{\pi\tau} d\tau\right| \tag{86}$$

$$= t\left|\int_{\omega_{\max}t}^{\omega_{\max}(t+T)} \frac{\sin\tau}{\pi\tau} d\tau\right| \tag{87}$$

$$= \frac{t}{\pi}\left|\int_{\omega_{\max}t}^{\omega_{\max}(t+T)} \frac{\sin\tau}{\tau} d\tau\right| \overset{(i)}{\leq} \frac{t}{\pi}\frac{2}{\omega_{\max}t} = \frac{2}{\pi\omega_{\max}} \tag{88}$$

Inequality (i) is due to Proposition D.2

*Bound on the fourth integral* $\int_{\omega_{\max}(t+T)}^{\infty} \Delta s\left(t - \frac{\tau}{\omega_{\max}}\right) \frac{\sin\tau}{\pi\tau} d\tau$*: When* $\tau \in (\omega_{\max}(t+T), \infty)$, we have $-\infty < t - \tau/\omega_{\max} < -T$. Then, $s\left(t - \frac{\tau}{\omega_{\max}}\right) = 0$.

$$\left|\int_{\omega_{\max}(t+T)}^{\infty} \Delta s\left(t - \frac{\tau}{\omega_{\max}}\right) \frac{\sin\tau}{\pi\tau} d\tau\right| = \left|\int_{\omega_{\max}(t+T)}^{\infty} (T-t)\frac{\sin\tau}{\tau} d\tau\right|$$

$$\overset{(i)}{\leq} \frac{(T-t)\times 2}{\omega_{\max}(t+T)\pi} \tag{89}$$

$$\leq \frac{2}{\pi\omega_{\max}}. \tag{90}$$

Inequality (i) is due to Proposition D.2 Hence,

$$|\widehat{s}_{\omega_{\max}}(t) - s(t)| \leq \frac{6 + 2\max\left\{1, \frac{T}{T+t}\right\}}{\pi\omega_{\max}} \tag{91}$$

The above result, together with $\text{sim}(q, x) = \sum_{k\in[K]} s(q[k] - x[k])$ and $\widehat{\text{sim}}_{\omega_{\max}}(q, x) = \sum_{k\in[K]} \widehat{s}_{\omega_{\max}}(q[k] - x[k])$ prove the Lemma. Q.E.D.

**Proposition D.2.** If $\alpha$ and $\beta$ have the same sign and $\alpha < \beta$, then we have:

$$\left|\int_{\alpha}^{\beta} \frac{\sin t}{\pi t} dt\right| \leq \begin{cases} \frac{2}{\pi\alpha} & \text{if } \beta > \alpha > 0 \\ -\frac{2}{\pi\beta} & \text{if } 0 > \beta > \alpha \end{cases} \tag{92}$$

**Proof** Note that $\frac{d}{dt}\frac{\cos t}{t} = -\frac{\sin t}{t} - \frac{\cos t}{t^2}$. Then we have:

$$\int_{\alpha}^{\beta} \frac{\sin t}{t} dt = -\int_{\alpha}^{\beta} \frac{d}{dt}\left[\frac{\cos t}{t}\right] dt - \int_{\alpha}^{\beta} \frac{\cos t}{t^2} dt \tag{93}$$

$$= \frac{\cos\alpha}{\alpha} - \frac{\cos\beta}{\beta} + \int_{\alpha}^{\beta} \frac{\cos t}{t^2} dt. \tag{94}$$

This gives us the following inequality:

$$\left|\int_{\alpha}^{\beta} \frac{\sin t}{t} dt\right| \leq \left|\frac{\cos\alpha}{\alpha} - \frac{\cos\beta}{\beta}\right| + \left|\int_{\alpha}^{\beta} \frac{\cos t}{t^2} dt\right| \tag{95}$$

$$\leq \frac{1}{|\alpha|} + \frac{1}{|\beta|} + \int_{\alpha}^{\beta} \frac{|\cos t|}{t^2} dt \tag{96}$$

$$\leq \frac{1}{|\alpha|} + \frac{1}{|\beta|} + \frac{1}{\alpha} - \frac{1}{\beta}$$

$$= \begin{cases} \frac{2}{\alpha} & \text{if } \beta > \alpha > 0 \\ -\frac{2}{\beta} & \text{if } 0 > \beta > \alpha \end{cases} \tag{97}$$

# E    Additional details about the experimental setup

## E.1    Dataset Generation

We obtain the MSWEB[1] and MSNBC[2] datasets from the UCI Machine Learning repository. Both of the datasets contain anonymized logs of real world user web activity. Each data item in MSWEB is a set of text snippets denoting areas of the website *www.microsoft.com* visited by an user within a specified time frame. Similarly, MSWEB consists of multi-sets denoting user page requests under various news categories at *www.msnbc.com*. Overall, we regard each data set as a collection of items, each item being a bag of words. For each data set, we sample a subset of items and designate them as queries, and the remaining items are designated as corpus items. The (binary) query-corpus relevance for MSWEB-1 and MSWEB-2 are governed by set containment, while for MSNBC-1 and MSNBC-2 we use multi-set (bag) containment. I.e., $x$ is relevant for $q$ iff $q \subseteq x$. To test the ability of FOURIERHASHNET to retrieve semantically similar items close to the gold items, we report not only on MAP based on gold labels but also scores of the top-10 candidates (Figure 3). The dataset characteristics are summarized in Table 7. We create datasets which differ greatly in terms of corpus size (10734 for MSWEB-1 and MSWEB, 110790 for MSNBC-1 and MSNBC-2), as well as span a range of average query selectivity between $1.7 \times 10^{-4}$ and $3.3 \times 10^{-3}$. We set aside 100 query graphs each for training and validation, and use the remaining 300 for testing.

| Dataset | $\|Q\|$ | $\|X\|$ | $\frac{\sum_{q \in Q x \in X} \text{rel}(q,x)]}{\|Q\|}$ | $\frac{\min_{q \in Q x \in X} \text{rel}(q,x)]}{\|Q\|}$ | $\frac{\max_{q \in Q x \in X} \text{rel}(q,x)]}{\|Q\|}$ | $\frac{\sum_{q \in Q x \in X} \text{rel}(q,x)]}{\|Q\|\|C\|}$ |
|---|---|---|---|---|---|---|
| MSWEB-1 | 500 | 10734 | 35.624 | 9 | 327 | 0.0033 |
| MSWEB-2 | 500 | 10734 | 20.392 | 9 | 49 | 0.0019 |
| MSNBC-1 | 500 | 110790 | 24.09 | 9 | 44 | 0.00022 |
| MSNBC-2 | 500 | 110790 | 19.78 | 9 | 34 | 0.00017 |

Table 7: Dataset statistics. From left to right: Datasets name, number of queries, number of corpus, the average number of relevant corpus items per query, the minimum num of relevnt corpus items per query, the maximum number of corpus items per query and the average query selectivity.

## E.2    Learning Representations for $q$ and $x$ for the baselines

During experiments for Section 4.2, in each of the baseline method (Cosine similarity, Dot product and Weighted Jaccard), we use the respective similarity scoring functions, and minimize a pairwise ranking loss on the gold relevance labels to learn the deep set network. We observed that the pairwise ranking loss performs better than the BCE loss for the baselines. The margin enabled pairwise ranking loss is specified as

$$\text{Loss} = \sum_{q \in Q} \sum_{\substack{x \in X_{q\checkmark} \\ x' \in X_{q\times}}} \left[\text{margin} + \text{sim}(q, x') - \text{sim}(q, x)\right]_{+}. \tag{98}$$

where sim is the choice of similarity scoring baseline, $X_{q\checkmark}, X_{q\times}$ are the set of relevant and irrelevant corpus items for the query $q$. We use the best performing margin among $\{1, 0.1\}$.

## E.3    Sampling from arbitrary distribution

One key component of FOURIERHASHNET is sampling $\boldsymbol{\omega}^{1\dots M} \sim p(\boldsymbol{\omega})$ . We have chosen to set $p(\omega) \propto |\text{Re}(S(\iota\omega))| + |\text{Im}(S(\iota\omega))|$, with the support set between $p(\omega)$ between $[-100, 100]$. The samples are drawn using Inverse Transform Sampling.

## E.4    Details about fourier transformation network

In our experiments, we generate $M = 10$ samples for $\boldsymbol{\omega}$. The neural networks $\phi_q$ and $\phi_x$ are linear layers which output 10 dimensional transformed Fourier representations $\boldsymbol{z}_q = \phi_q(\boldsymbol{F}_q(\iota\boldsymbol{\omega}^{1\dots M}))$ and $\boldsymbol{z}_x = \phi_x(\boldsymbol{F}_x(\iota\boldsymbol{\omega}^{1\dots M}))$. These are trained using the BCE loss specified in Eq. (12).

---

[1] https://archive.ics.uci.edu/ml/machine-learning-databases/msweb-mld
[2] https://archive.ics.uci.edu/ml/machine-learning-databases/msnbc-mld/

### E.5 Details about hashcode generation network

We use the same hashcode training procedure for FOURIERHASHNET, as well as the DP-RH and RH baselines. In all three cases, we generate 64 dimensional hashcodes. For FOURIERHASHNET, the random hyperplanes $W$ are trained on 10 dimensional trained Fourier representations $z_q, z_x$. For RH we use the original embeddings $q$ and $x$. For DP-RH, we use the augmented embeddings $g(q) = \text{sign}(W[0, q/||q||])$ and $h(x) = \text{sign}(W[\sqrt{T^2 - ||x||^2}, x])$.

### E.6 AP and MAP measurements

Suppose a query $q$ is associated with $N_{q\oplus}$ relevant corpus items (as judged by humans). Suppose the system provides a ranking over all $N$ corpus items, and the relevant items occur at ranks $r_1, \ldots, r_{N_{q\oplus}}$. Then AP for query $q$ is defined as $(1/N_{q\oplus}) \sum_{i=1}^{N_{q\oplus}} (i/r_i)$. This is because, up to position $r_i$, we have seen $i$ relevant items, which means we can shorthand $i/r_i$ as prec@$i$ (precision at $i$). We can rewrite the sum as $\frac{1}{N_{q\oplus}} \sum_{r=1}^{N} \text{prec}@r \times \text{rel}@r$, where $N$ is the size of the whole corpus, and $\text{rel}@r$ is the (0/1) relevance of the item at rank $r$. In case the retrieval algorithm does not assess all $N$ corpus items, but stops with the best $L$ hash buckets, which contain, say, $N_q'$ items, we should use the following formula for AP: $\frac{1}{N_{q\oplus}} \sum_{r=1}^{N_q'} \text{prec}@r \times \text{rel}@r$. Note that we should still divide by $N_{q\oplus}$, otherwise an algorithm that maps the query to a densely relevant but small bin, which fails to retrieve most relevant items, might be rewarded in an unfair manner.

### E.7 Top-10 score measurement

In Appendix F, we provide additional experiments where we compare FOURIERHASHNET with all the baselines not only in terms of MAP, but also in terms of the Top-10 score. We use the sum of Top-10 scores normalized in $[0, 1]$ via the sigmoid transformation used in Eq. (15): Top-10$(q) = \sum_{x \in \text{Top-10}(N_q')} \sigma(-d(q, x))$. Any hashing protocol is expected to retrieve the corpus items, which have the highest similarity scores with respect to any given query. The Top-10 score evaluates it independently of how the retrieved items match with true relevant items. Therefore, the Top-10 scores provide an evaluation mechanism that is independent of the gold relevance labels and solely relies on the scores dictated by the trained embeddings. This offers a valuable means of assessing performance without being influenced by subjective human judgments.

### E.8 Licenses

We utilize a publicly available pre-trained sentence transformer model [41], which is licensed under the Apache License 2.0. Additionally, we employ the DrHash toolkit [58] for various implementations of the baseline Weighted Minhash (WMH) algorithms. The DrHash toolkit is publicly available under the MIT License. We duly acknowledge the original authors of the baseline methods in our citations.

# F Additional experiments

## F.1 Applying baseline LSHs on hinge distance guided embeddings

In continuation of the results reported in Figure 3, in Figure 8, we present the complete view as well as the zoomed versions of the trade-off curves for all datasets.

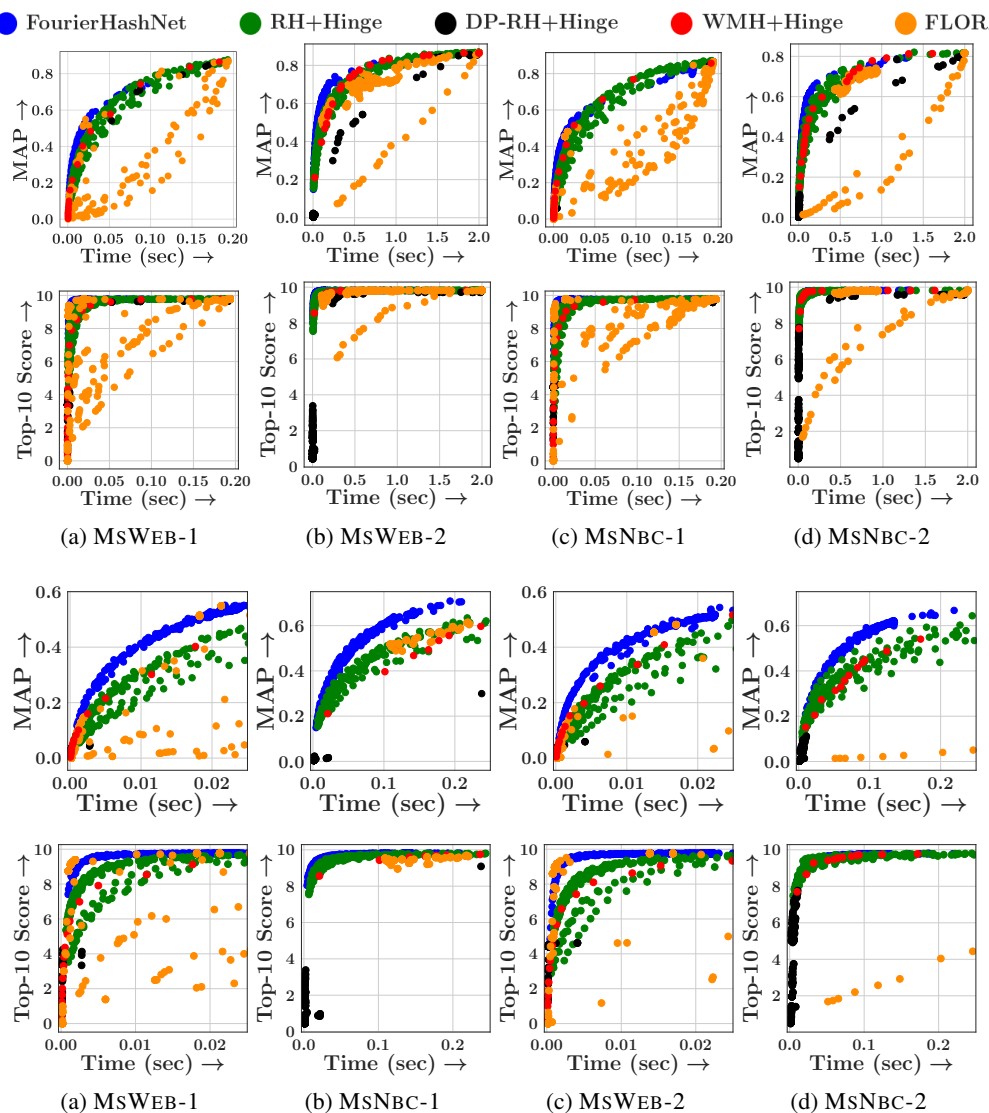

Figure 8: Trade-off between average query time and accuracy (MAP and Top-10 scores) for MsWEB and MsNBC datasets (First two rows: complete view across full time axis, last two rows: Zoomed version of first two rows until the average query time there is $\geq$ 10X speedup compared to exhaustive search). We apply different LSH methods on hinge distance guided embeddings similar to Figure 3, then use the hinge distance to finally rank the retrieved items.

Beyond the observations made in Figure 3, we make the following additional observations.
**(1) The complete view for both Top-10 score and MAP score, clearly demonstrates FLORA's high sensitivity to hyperparameter tuning. FLORA is seen to have the highest variance in scores for any given time budget across all the baselines. In terms of Top-10 score, while FLORA is marginally ahead of FOURIERHASHNET in a few instances in the MsWEB datasets, it is significantly outperformed by FOURIERHASHNET in the MsNBC datasets. This may be due to the significantly higher average query selectivity in the MsNBC datasets.**
**(2)** In terms of Top-10 score, FOURIERHASHNET achieves the maximum possible value $4\times$ faster than the nearest competitor RH+Hinge, in MsWEB-1 and MsWEB-2. In terms of MAP score,

RH+Hinge and WMH+Hinge achieve a maximum MAP of 0.5 in MSWEB-1 and MSWEB-2, and 0.62 in MSNBC-1 and MSNBC-2. However, FOURIERHASHNET achieves the same MAP values 1.33× faster in MSWEB-1 and MSWEB-2, and 2× faster in MSNBC-1 and MSNBC-2.
**(3)** Interestingly, the gap between FOURIERHASHNET and RH+Hinge seems to widen for MSWEB-2, when compared to MSWEB-1. This is possibly due to the presence of several queries in MSWEB-1, which have ≥300 relevant corpus items. This affords RH+Hinge a greater opportunity to fetch high scoring items, which is not the case in MSWEB-2.

## F.2 Ablation study on hashcode training

In continuation of the results reported in Figure 5, in Figure 9, we present the complete view as well as the zoomed versions of the trade-off curves for all datasets.

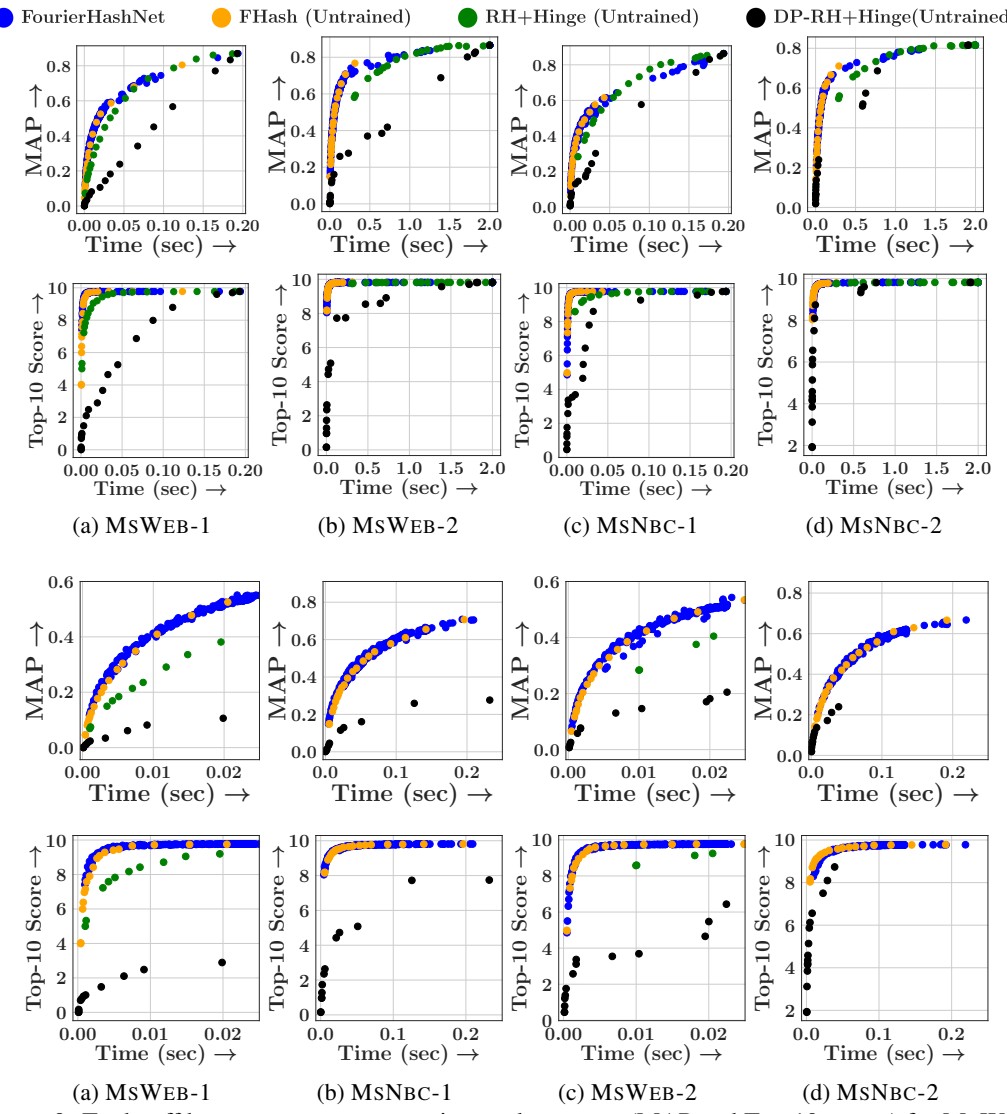

Figure 9: Trade-off between average query time and accuracy (MAP and Top-10 scores) for MSWEB and MSNBC datasets (First two rows: complete view across full time axis, last two rows: Zoomed version of first two rows until the average query time there is ≥ 10X speedup compared to exhaustive search). We compare FHASH (UNTRAINED) against the untrained versions of RH+Hinge and DP-RH+Hinge, as well as against FOURIERHASHNET.

Beyond the observations made in Figure 5, we make the following additional observations.
**(1)** In terms of both Top-10 score and MAP score, FHASH (UNTRAINED) clearly outperforms both RH+Hinge and DP-RH+Hinge, across all four datasets. This strongly highlights the advantage of our

Fourier feature generation method.

**(2)** In every setup, FOURIERHASHNET enables a wider range of options for accuracy score vs retrieval time trade-off.

### F.3  Ablation study on collision minimizer (14)

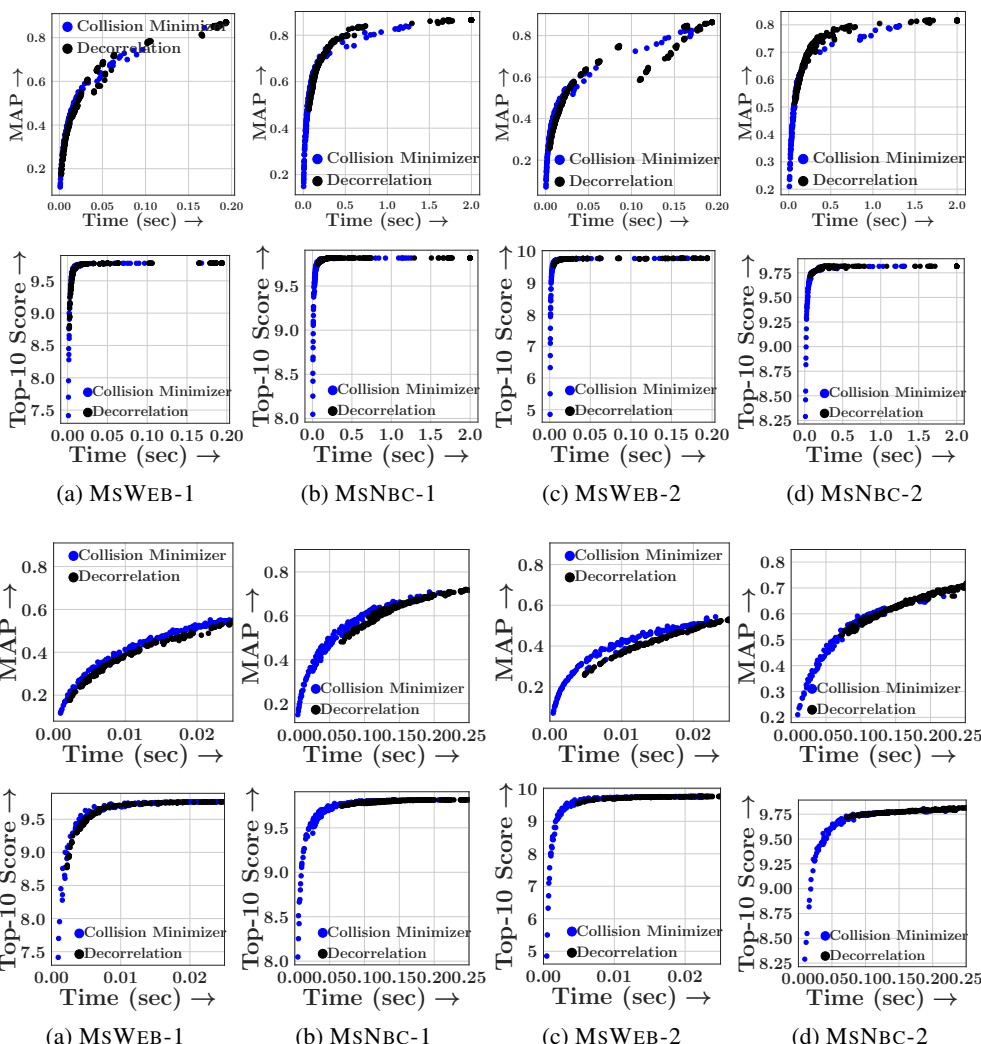

Figure 10: Trade-off between mean query time and accuracy (MAP and Top-10 scores) for MSWEB and MSNBC datasets (First two rows: complete view across full time axis, last two rows: Zoomed version of first two rows until the mean query time there is $\geq 10X$ speedup compared to exhaustive search). We compare our hashcode training loss $\text{loss}(Q, X \mid \boldsymbol{W})$ (14), against a variant which replaces the collision minimizer component $\Delta_1$ with a decorrelation loss $\sum_{x \neq y} |\tanh(\boldsymbol{W}\boldsymbol{z}_x)^\top \tanh(\boldsymbol{W}\boldsymbol{z}_y)|$.

In continuation of the results reported in Figure 6, we present the complete view as well as the zoomed versions of the trade-off curves for all datasets. Beyond the observations made in Figure 6, we make the following additional observations.

**(1)** In MSWEB-2, the alternative variant shows a sudden plunge in MAP performance trade-off at around 0.1 seconds. This type of discontinuous drop is not observed in any of our cases.

**(2)** In MSWEB-1, there is a variation of 0.1 MAP at around 0.05 seconds. Such high variability is not observed for any of our trade-off curves.

### F.4  Identifying best performing Weighted Minhash algorithm for our datasets

For implementation of baseline Weighted Minhash (WMH) algorithm, we use the best performing WMH implementation available in the DrHash toolkit [58] to obtain the hashcodes. We comapare

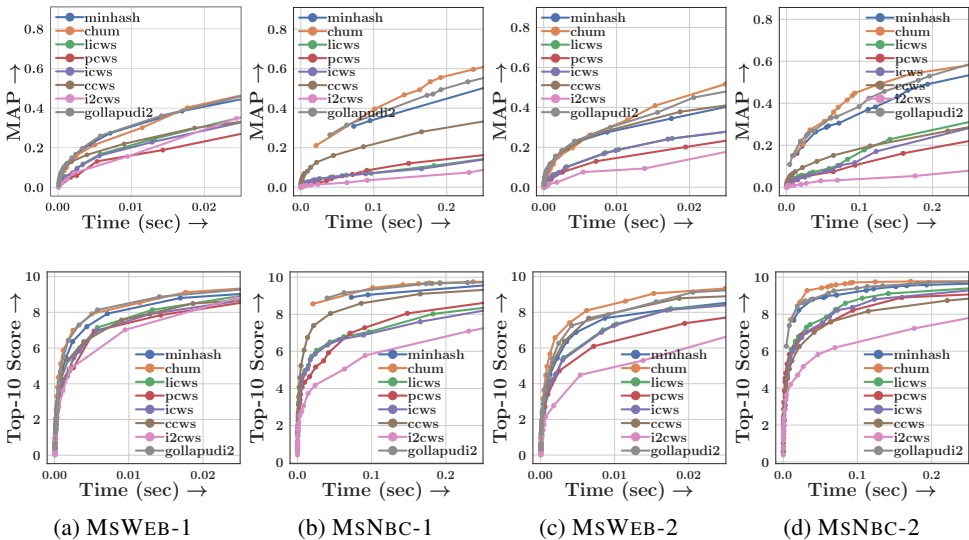

Figure 11: We compare performance of Weighted Minhash variations, in terms of trade-off between mean query time and accuracy (MAP and Top-10 scores) for MSWEB and MSNBC datasets, until the query time there is $\geq$10X speedup compared to exhaustive search.

across the 8 available baselines in the toolkit: minhash [7], chum [12], icws [20], pcws [56], licws [28], ccws [55], i2cws [57] and gollapudi2 [16].

We make the following observations.

(1) Across all four of our datasets, for both MAP and Top-10 score, the top 3 performers are minhash, gollapudi2 and chum. The remaining algorithms are often significantly worse in performance, as can be seen for MAP in all 4 datasets and for Top-10 in MSNBC-2.

(2) Among the top 3 performers, chum is seen to be the best perform in terms for both MAP and Top-10 score, in MSWEB-2, MSNBC-1 and MSNBC-2. In MSWEB-1, chum is tied with minhash and gollapudi2 for the top position.

Driven by these observations, we choose chum as the representative baseline for WMH in our experiments.

## F.5 Effect of $M$, number of samples of $\omega$ on FOURIERHASHNET performance

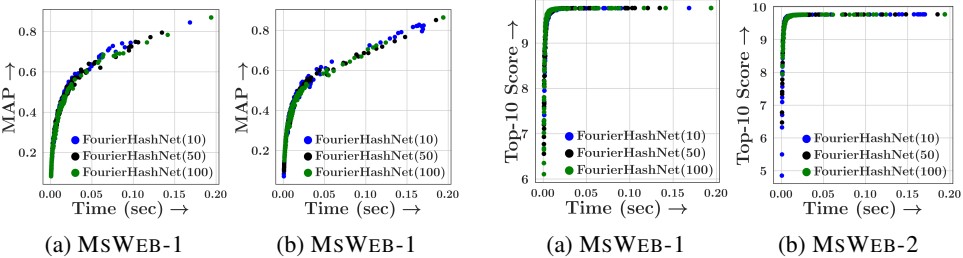

Figure 12: Effect of $M$, number of $\omega$ samples, on the trade-off between mean query time and accuracy (MAP and Top-10 scores) for MSWEB datasets.

Here we check the impact of varying the number of samples ($M$) for $\omega$. We consider three different values of $M$, *i.e.*, 10, 50 and 100, for generating the fourier features $\boldsymbol{F}_q(\iota\boldsymbol{\omega}^{1..M})$ and $\boldsymbol{F}_x(\iota\boldsymbol{\omega}^{1..M})$ which are then fed into the neural networks $\phi_q$ and $\phi_x$ for learning the transformed Fourier representations $\boldsymbol{z}_q$ and $\boldsymbol{z}_x$, using the BCE loss specified in Eq. (12). Finally, we train the random hyperplanes $\boldsymbol{W}$ and check the MAP and Top-10 score performances for the three variations - FOURIERHASHNET(10), FOURIERHASHNET(50) and FOURIERHASHNET(100). We observe that the final performance trade-off of both MAP and Top-10 scores, remains roughly the same across all three variants. This shows that trainable Fourier transformation is able to compensate for the quality of Monte Carlo approximations affected by the number of $\omega$ samples $M$.

Next, we investigate how well the MC estimates of the Fourier features approximate the dominance similarity function $\text{sim}(q, x)$. Here, we set the dimension of $\boldsymbol{q}$ and $\boldsymbol{x}$ as $K = 1$. We set $T = 20$ and we sample $\boldsymbol{q}, \boldsymbol{x} \sim \text{Unif}[-20, 20]$. Finally, we compute $\widehat{\text{sim}}_M(q, x) = \frac{||\boldsymbol{F}_\bullet(\iota\boldsymbol{\omega}^{1..M})||^2}{M} \cos(\boldsymbol{F}_q(\iota\boldsymbol{\omega}^{1..M}), \boldsymbol{F}_x(\iota\boldsymbol{\omega}^{1..M}))$ and measure the variation of $\epsilon_{\text{sim}} = \mathbb{E}_{\boldsymbol{q},\boldsymbol{x}\sim\text{Unif}[-20,20]}[||\text{sim}(q, x) - \widehat{\text{sim}}_M(q, x)||]$ with $M$, the number of samples of $\omega$. Figure 13 summarizes the results, which shows as $M$ increases, $\epsilon_{\text{sim}}$ decreases.

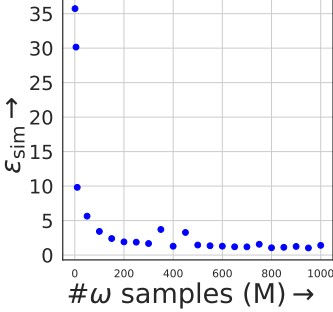

Figure 13: Variation of $\epsilon_{\text{sim}} = \mathbb{E}_{\boldsymbol{q},\boldsymbol{x}\sim\text{Unif}[-20,20]}[||\text{sim}(q, x) - \widehat{\text{sim}}_M(q, x)||]$ with $M$, the number of $\omega$ samples.

### F.6 Applying baseline LSHs on hinge distance guided embeddings with noisy labels

In certain applications, the accuracy of ground truth labels can be compromised by noise or subjective human judgments of relevance. We evaluate the performance of FOURIERHASHNET and the baselines in a noisy label setup to test its robustness.

Starting with the hinge distance guided embeddings, we initially rank the corpus items based on their dominance similarity scores. Subsequently, we intentionally flip the labels of the bottom-ranking 10% of positive labels to negative, and an equal number of highest ranked negatively labeled items to positive. This simulation reflects a plausible scenario since the lowest ranked positive items and the highest ranked negative items are particularly susceptible to misclassification in real-world settings. Furthermore, this approach ensures that the average query selectivity for each dataset remains unchanged.

As before, we apply different the LSH methods on hinge distance guided embeddings, *viz*, RH+Hinge, DP-RH+Hinge, WMH+Hinge, FLORA and FOURIERHASHNET; and, then use the hinge distance to finally rank the retrieved items.

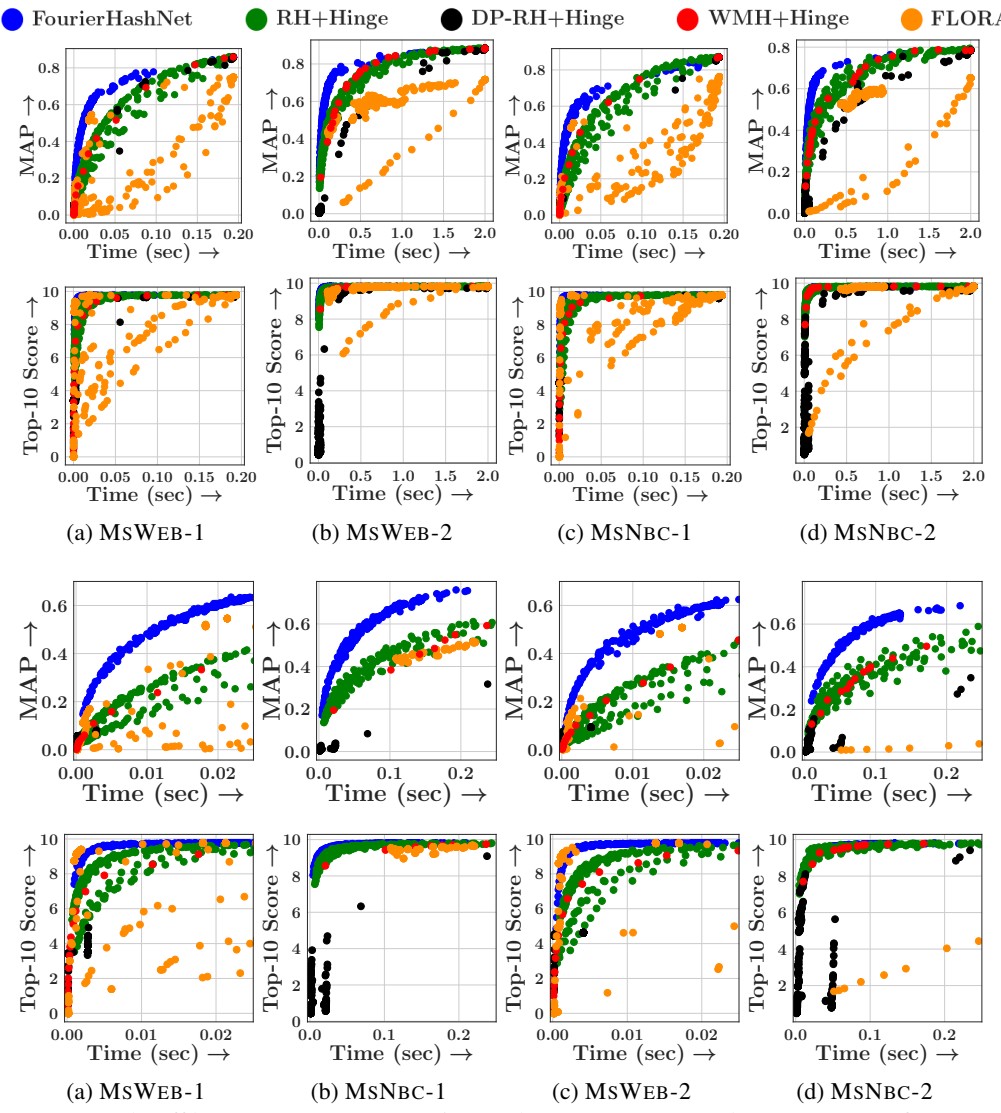

Figure 14: Trade-off between average query time and accuracy (MAP and Top-10 scores) for MSWEB and MSNBC datasets (First two rows: complete view across full time axis, last two rows: Zoomed version of first two rows until the average query time there is ≥ 10X speedup compared to exhaustive search). We apply different LSH methods on hinge distance guided embeddings similar to Figure 8 , and then use hinge distance to rank retrieved items. Evaluations are conducted using noisy labels.

We make the following observations.

**(1)** In terms of MAP score, FOURIERHASHNET continues to outperform all other baselines across all four datasets. Furthermore, when comparing the performance in the presence of noise, as depicted in Figure 14, to the corresponding results obtained in the noiseless setting illustrated in Figure 8, we observe that FOURIERHASHNET outperforms all its competitors by a significantly higher margin in the presence of noise.

**(2)** In terms of Top-10 score, we note that the results in Figure 14 for the noisy setup are identical to the results presented in the noiseless setting shown in Figure 8. This observation indicates that the evaluation based on Top-10 score is unaffected by label noise. This supports the argument made in Appendix E.7 that Top-10 score evaluation enables a more subjective assessment of performance, focusing on the quality of the embeddings themselves.

## F.7 Comparison with graph based indexing methods

Here, we compare the performance of FOURIERHASHNET with a representative graph based indexing method. We use the Hierarchical Navigable Small Worlds (HNSW) implementation of Hnswlib [33]. We extend the `SpaceInterface` class of `nmslib/Hnswlib` to implement HNSW for hinge distance. In order to track the number of distance computations performed by HNSW during retrieval, we used a counter inside `fstdistfunc_`. We count the number of distance computations as a surrogate for real time, to avoid non-determinism in measurements and low-level implementation differences. We search across different values of M, ef and ef_construction, and track the number of distance computations against corresponding MAP values.

| #calls to `fstdistfunc_` | MAP | Method |
|---|---|---|
| 839 | 0.17 | HNSW |
| 1162 | 0.53 | FOURIERHASHNET |
| 1549 | 0.43 | HNSW |
| 1668 | 0.51 | HNSW |
| 1846 | 0.58 | FOURIERHASHNET |
| 2578 | 0.59 | HNSW |
| 3529 | 0.71 | HNSW |
| 3926 | 0.72 | FOURIERHASHNET |
| 4773 | 0.77 | HNSW |

Table 15: Number of distance computations and the corresponding MAP values on MSWEB-1 for HNSW and FOURIERHASHNET.

Table presents our study on MSWEB-1 dataset with 10734 corpus items. We observe that FOURIERHASHNET LSH has an edge over HNSW in the regime of fewer distance computations, with a MAP of 0.53 using 1162 distance computations. However, when allowed more distance computations, HNSW outperforms FOURIERHASHNET with a MAP of 0.71 in 3528 computations, and a MAP of 0.77 in 4773 computations.

