# OpenReview forum: "Locality Sensitive Hashing in Fourier Frequency Domain For Soft Set Containment Search"
_NeurIPS.cc/2023/Conference — NeurIPS 2023 spotlight_

### Official Review · Reviewer_wHYk · 2023-07-05

**Soundness:** 3 good
**Presentation:** 3 good
**Contribution:** 3 good
**Rating:** 6
**Confidence:** 4

**Summary:**

This paper presents a novel approach, called FourierHashNet, for fast soft set containment search. The key idea is to extend set containment to soft set containment by representing query and document elements as embedded representations, instead of atomic IDs. The authors propose a dominance similarity measure based on hinge distance and transform it into the frequency domain using a Fourier transform. This allows for the efficient use of traditional LSH techniques.


**Strengths:**

Strength:

1. The significance of the proposed asymmetric dominance similarity measure is emphasized, indicating its critical role in the targeted applications. By transforming the dominance similarity measure into the frequency domain, the authors enable the utilization of traditional LSH methods. This approach not only enhances retrieval efficiency but also demonstrates a better trade-off between query time and retrieval quality.

2. This paper is well-written. It effectively communicates the technical aspects of the proposed method, including the use of the Fourier transform and the learning of data-sensitive hash codes.


**Weaknesses:**

NA

**Questions:**

Is it possible to discuss the potential of FourierHashNet in the similarity search of general kernel measure?

**Limitations:**

Yes, the authors adequately addressed the limitations

---

> ### Author Rebuttal · Authors · 2023-08-09
>
> The authors thank reviewer wHYk for the positive feedback.
> In the general response,  we delve into a broader application context of FourierHashNet, particularly for similarity search involving shift invariant functions.
> Please reach out to us if you have further inquiries or points for clarification.

---

### Official Review · Reviewer_AbHb · 2023-07-06

**Soundness:** 3 good
**Presentation:** 2 fair
**Contribution:** 4 excellent
**Rating:** 6
**Confidence:** 4

**Summary:**

Locality-sensitive hash (LSH) functions are mappings from sets of queries and documents to "buckets," in such a way that similar queries and documents are assigned to the same bucket. This paper proposes an asymmetric LSH where the notion of similarity is related to the hinge distance (called dominance similarity, based on the element-wise inequalities between components of two vectors). The method conceptually follows three steps: truncate, transform, and sample. First, the dominance similarity is truncated to a bounded region of the input domain; this is done so that the problem is tractable. Then, the similarity is Fourier-transformed into a form where the contributions from different vectors separate into an inner product. This form is used to sample Fourier representations that are used to compute the LSH in practice.

The paper also contains experiments comparing the proposed ALSH with other LSH algorithms on two web log datasets, where the task is to perform vector search under the hinge distance. FourierHashNet outperforms these methods by a sizable margin. To further improve practical performance, the paper proposes a learnable version of the algorithm and conducts ablations to investigate its effect.

**Post rebuttal:** The authors provide a general framework for ALSH, of which the hinge distance LSH is a specific instance. This substantially extends the utility and impact of the method over the original proposal.


**Strengths:**

**S1: Important topic.** The problem of designing an asymmetric LSH with an exotic collision probability is useful and interesting. There is not much work in this area, but there are several emerging LSH applications (not just search) where this capability will be helpful. For example, the [LGD sampler [NeurIPS 2019]](https://proceedings.neurips.cc/paper_files/paper/2019/file/a1e865a9b1065392ed6035d8ccd072d9-Paper.pdf) uses LSH functions that are sensitive to classification loss functions. Some of these are non-symmetric, so new ways to construct asymmetric LSH can correspond to faster optimization routines for more problems.
The efficient kernel-matrix multiplication algorithm of Backurs et. al. [[ICML 2021]](https://arxiv.org/abs/2102.08341) is based on LSH and currently only works for symmetric kernels - asymmetric LSH could substantially extend this framework.
An asymmetric LSH was recently developed for the linear regression/classification loss and was used to learn differentially-private classifiers [[CCS 2021]](https://dl.acm.org/doi/abs/10.1145/3460120.3485255); new ALSH functions will also directly expand the usefulness of this framework.
Therefore, ALSH is an increasingly important topic with the potential to impact search, ranking, optimization, numerical linear algebra, differential privacy, and likely other areas.

**S1: Novelty.** This paper attacks a technically difficult problem (ALSH for the hinge distance) and introduces some new ALSH techniques to do the design. I checked the math and the results seem to be correct. Note that I did not check details (e.g. the algebra involved in going from (7) to (8), or the derivation of (5) in the appendix).


**Weaknesses:**

**W1: Presentation.** The presentation and flow of the argument can be improved. Here are some specific examples:
- The "i" used to denote sqrt(-1) should probably be the standard italic i (with a dot) rather than the symbol \i.
- It may be more correct to refer to the "hinge distance" as the "hinge quasimetric" or the "hinge divergence" because it is not strictly speaking a metric (it obeys all properties except symmetry).
- On line 177, consider mentioning that Equation (3) is just the Fourier Transform of -d(q,x) from Equation (1). It took me a moment to determine why (3) was written as the sum rather than as K iterated integrals.
- Equation (4) could be introduced as a "clamped" or "truncated" version of -d(q,x), to make its definition more clear to the reader.
- The "Fence Sitting" and "bit balance" losses don't seem particularly well-known (the only reference I could find was in "Adversarial Permutation Guided Node Representations for Link Prediction" in [AAAI 2021](https://arxiv.org/abs/2012.08974) - this paper should probably be cited). Therefore, these losses could possibly use some more explanation.
- The results figures for the experiments would benefit from better layout to make the graphics larger (e.g. using subplots with tight_layout or similar).
- Several components seem out of place. For example, the formal definitions of LSH and ALSH are introduced but are not used to prove that the hash function is indeed an ALSH. Observations about symmetric LSH (e.g. the paper by Chierichetti and Kumar) are mentioned throughout the development of the asymmetric LSH in Section 3, and it might be clearer to collect all these arguments in Section 2. The sampling algorithm is fairly involved and probably merits its own section.

**W2: Impact.** The main contribution of the paper is an asymmetric LSH function for a very specific similarity function. This paper could have an impact on the LSH algorithm area and on any area that uses the hinge distance for set search. However, I have some concerns about the applicability of the ideas outside of what seems to be a fairly niche search problem.

- **On LSH.** As mentioned above, there are several problems that can benefit from asymmetric LSH. However, it is hard to see which contributions from this paper might be more broadly applicable. It would be much more interesting (and appeal to a wider audience) if this algorithm can be generalized into a broader "truncate - transform - sample" template. Such a template might apply directly to other applications. It is also fairly essential to demonstrate that this is, indeed, an ALSH (that provably satisfies Definition 2.2).
- **On set-search.** The community seems to be moving away from representing sets as a single vector and toward approaches that examine the pairwise similarity between all elements in the set (e.g. [ColBERT](https://arxiv.org/pdf/2004.12832.pdf), [ColBERT v2](https://arxiv.org/abs/2112.01488), [PLAID](https://arxiv.org/abs/2205.09707), several follow-ups). This approach was originally motivated by the comparatively high cost of performing transformer inference but is now [doing very well](https://arxiv.org/abs/2212.01340) on tasks like passage ranking. Many of the papers cited as applications are more than five years old (pre-2019), so it is not clear whether search with the hinge distance is still a pressing problem.

**W3: Experiments.** The baselines in the experiments are not very competitive, leading to a limited evaluation. LSH is a classical, standard technique but it is no longer state-of-the-art for most search/ranking problems. Other approaches (like FAISS-IVF or HNSW) have taken over as the search algorithm of choice. These methods are also compatible with the hinge distance (though, this does require use of the C++ APIs rather than the simpler Python API). They also tend to be an order of magnitude faster than LSH, according to well-established benchmarks ([ann-benchmarks](https://ann-benchmarks.com), see also the NeurIPS 21 competition [big-ann-benchmarks](https://big-ann-benchmarks.com)).

**Questions:**

I like this line of research and would be willing to raise my score if some of the following items can be addressed. In particular, I'm looking for evidence that the hinge distance is an important bottleneck and for evidence that the new techniques developed in this paper could be useful more broadly.

- Is it possible to generalize the sampling process (bottom of page 5) into a generic algorithm with theoretical guarantees (e.g. on the expectation and possibly variance of the estimate)? This would go a long way towards addressing W2, as it would be a result that other works could build off of.

- Is element-wise vector inequality the current state-of-the-art for the applications mentioned in the paper? ColBERT (which examines all pairwise relationships between the two sets) and follow-ups have shown very good results in natural language applications, and determinental point processes are SOTA in many market basket analysis applications (e.g. "Learning Nonsymmetric Determinantal Point Processes" at [NeurIPS 2019](https://proceedings.neurips.cc/paper_files/paper/2019/file/cae82d4350cc23aca7fc9ae38dab38ab-Paper.pdf)). I am less familiar with the knowledge graphs and subgraph isomorphism applications mentioned in the appendix so it might be possible that the problem is very strongly motivated by these areas. I did take a quick look at the references but I did not find this search problem mentioned.

- Can this algorithm handle box embeddings (perhaps with some modifications to the algorithm / the embeddings)? Search over box embeddings is a known bottleneck and is one reason why, despite modeling advantages, they have not replaced angular-similarity embeddings in major industrial recommendation pipelines.

- Is it possible to prove that this scheme is an ALSH (i.e. satisfies Defintion 2.2) for the hinge distance? Due to the truncation in Equation (4), it might be necessary to restrict the domain over which we expect Definition 2.2 to hold (but this would be fine and would not affect the quality of the theoretical result).

- Just to clarify - what are the "gold" and "silver" instances on page 6? Is this the same as the gold relevance labels from the datasets (page 7)?


**Limitations:**

Yes - no issues.

---

> ### Author Rebuttal · Authors · 2023-08-09
>
> We thank reviewer AbHb for the spectacular feedback.
> > Proof that FourierHashNet is an ALSH
>
> Note that $p(\omega^j_k)\propto|Re(S(\omega _k ^{j}))|+|Im(S(\omega _k ^{j}))|$. Let $I$ be the proportionality constant.  Assume $\mathrm{sim}(q,x)>s _m>0$ and  $\cos^{-1}$ is $L _{\cos}$-Lipschitz.
>
> We have: $||\pmb{F} _q(\pmb{\omega}^{1...M})|| _2^2=||\pmb{F} _x(\pmb{\omega}^{1...M})|| _2^2= \sum _{i\in[M],k\in[K]}\frac{|Re(S(\omega\_k^j))|+|Im(S(\omega\_k^j))|}{p(\omega^j _k)}=MKI$.
>
> We use $\Pr _{g,h}[g(q)=h(x)]=\mathbb{E} _{\pmb{\omega^{j}}}[\Pr _{g,h}[g(q)=h(x)|\pmb{\omega}^j]]$ to write
> \begin{align}
> &\Pr _{g,h}[g(q)=h(x)|\pmb{\omega}^i]=1-\frac{1}{\pi}\cos^{-1}\bigg(\frac{\pmb{F} _q(\pmb{\omega}^{1...M})^{\top}\pmb{F} _x(\pmb{\omega}^{1...M})}{||\pmb{F} _q(\pmb{\omega}^{1...M})||\ ||\pmb{F} _x(\pmb{\omega}^{1...M})||}\bigg) \\\\
> &=1-\frac{1}{\pi}\cos^{-1}\bigg(\frac{\operatorname{sim}(q,x)}{KI}\bigg) \\\\
> &-\frac{1}{\pi}\cos^{-1}\bigg(\frac{\pmb{F} _q(\pmb{\omega}^{1...M})^{\top}\pmb{F} _x(\pmb{\omega}^{1...M})}{MKI}\bigg)
> +\frac{1}{\pi}\cos^{-1}\bigg(\int _{\omega}\frac{\pmb{F} _q(\pmb{\omega})^{\top}\pmb{F} _x(\pmb{\omega})p(\pmb{\omega}) d \pmb{\omega}}{KI}\bigg)---(A)
> \end{align}
> Note
> \begin{align}
> &\mathbb{E}\bigg[-\cos^{-1}\bigg(\frac{\pmb{F}_q(\pmb{\omega}^{1...M})^{\top}\pmb{F}_x(\pmb{\omega}^{1...M})}{MKI}\bigg)
> +\cos^{-1}\bigg(\int _{\omega}\frac{\pmb{F}_q(\pmb{\omega})^{\top}\pmb{F}_x(\pmb{\omega})p(\pmb{\omega}) d \pmb{\omega}}{KI}\bigg) \bigg] \\\\
> &\le\frac{L _{\cos}}{KI}\mathbb{E}\bigg|\sum _{j\in[M]}\pmb{F}_q(\pmb{\omega}^{j})^{\top}\pmb{F}_x(\pmb{\omega}^{j})/M-\int _{\omega} \pmb{F} _q(\pmb{\omega})^{\top}\pmb{F} _x(\pmb{\omega})p(\pmb{\omega}) d \pmb{\omega}\bigg|
> \\\\
> &=\frac{L _{\cos}}{KI\sqrt{M}}\sqrt{K}\bigg(\text{Var}\bigg[\pmb{F}_q(\omega_k)^{\top}\pmb{F}_x(\omega_k )\bigg]\bigg)^{1/2}\le\frac{L _{\cos}}{\sqrt{KM}}---(B)
> \end{align}
> The last inequality follows from a bound on variance via
> \begin{align}
> &\pmb{F}_q(\omega_k)^{\top}\pmb{F}_x(\omega_k)\\\\
> &=\pmb{S}_q(\omega_k)^{\top}\pmb{S}_x(\omega_k)/p(\omega_k)\quad(\text{Eq 9 in paper})\\\\
> &=\frac{Re[\pmb{S}(\omega)]\cos\omega_k(q[k]-x[k])-Im[\pmb{S}(\omega)]\sin\omega_k(q[k]-x[k]))}{(1/I)[|Re[\pmb{S}(\omega)]|+ |Im[\pmb{S}(\omega)]|]}\\\\
> &\le I\frac{|Re[\pmb{S}(\omega_k)]|+|Im[\pmb{S}(\omega_k)]|}{|Re[\pmb{S}(\omega)]|+|Im[\pmb{S}(\omega)]|}=I
> \end{align}
>
> Putting (B) into (A), we have
> \begin{align}
> \mathbb{E}[\Pr\_{g,h}[g(q)=h(x)|\pmb{\omega}^{j}]]\le p_2=1-\frac{1}{\pi}\cos^{-1}\bigg(\frac{\operatorname{sim}(q,x)}{KI}\bigg)+ L_{\cos}/\pi\sqrt{KM}
> \end{align}
> Similarly, we have:
> \begin{align}
> \mathbb{E}[\Pr\_{g,h}[g(q)=h(x)|\pmb{\omega}^{j}]]\ge p_1=1-\frac{1}{\pi}\cos^{-1}\bigg(\frac{\operatorname{sim}(q,x)}{KI}\bigg)- L_{\cos}/\pi\sqrt{KM}
> \end{align}
>
> If we ensure
> \begin{align}
> M>\frac{4L _{\cos}^2}{K\left[\cos^{-1}\big(\frac{c\cdot s _m}{KI}\big)-\cos^{-1}\left(\frac{s _m}{KI}\right)\right]^2},
> \end{align}
> then we have $p_1>p_2$. This satisfies the condition for ALSH.
>
> > *W3: Experiments. ColBERT/FAISS/HNSW vs A/LSH and FourierHashnet*
>
> FourierHashNet beats FLORA (Fig 3 in main), which is a very recent (2023) neural retrieval model that outperforms HNSW.
>
> In response to your highlighting FAISS-IVF, we compare the retrieval performance of FAISS-IVF variants against FourierHashNet (Fig. 14 and 15 of the rebuttal PDF). FAISS-IVF retrieval suffers because its quantizers, that assign vectors to the Voronoi cells, rely on a metric like L2 or IP, which are unsuitable for asymmetric hinge distance. Also, IVF-style retrieval may fall short in other applications. We will incorporate the following discussion into the manuscript.
>
> **Textual entailment** (SNLI, MNLI) is usually presented as *classification*: (text_a, text_b) → {implies, contradicts, none} etc. The most accurate methods inject ([CLS], text_a, [SEP], text_b) into an early-cross-interaction transformer, and predict the label from the output [CLS] embedding.  The associated *retrieval* problem is, given text_b and a large corpus, quickly find top-K corpus text_a that are most likely to imply text_b. Early-cross-interaction precludes effective indexing/hashing. Lai+Hockenmaier propose asymmetric late interaction: separately obtain embeddings of text_a and text_b, and force the former to elementwise dominate the latter (by fine-tuning the transformer). Thanks to our work, dominance distance is now hashable.
>
> **Box embeddings** and **order embeddings** are used to model type hierarchies in knowledge graphs (KGs) and object hierarchies in images. If type t1 is subtype-of t2, we expect their order embeddings to follow elementwise dominance, or the box embeddings of t1 to be contained in that of t1. They can be used in our setup; see global response.
>
> > *In set-search, elementwise cross-interaction better than single set embeddings?*
>
> For applications other than retrieving soft supersets of the query set (as is the case in ColBERT-type passage retrieval) ColBERT and follow-ups may not be the best choice:
> * ColBERT uses a symmetric term-to-term similarity, which is inadequate for detecting subgraph isomorphism. unlike hinge distance — see Table 16 in rebuttal PDF.
> * If we are looking for subsets (rather than supersets) of the query, then our hinge distance continues to work. But ColBERT can now assign multiple query atoms to the same passage atom.
> * If the query (text or graph) is large, ColBERT (using IVF) will still hit many FAISS clusters and thus amass a large number of corpus items to score. This can be controlled better using ALSH in Fourier space.
>
> For these reasons we believe ALSH for hinge distance remains very strongly motivated.
>
> >*"gold" and "silver"*
>
> Instances with gold labels means ground-truth relevance labels.  After training with gold labels, silver labels are predicted from hinge distances. These are used to train the hashing protocol.
>
> >*broader application of "truncate - transform - sample" template.*
>
> Please refer to general response.

---

> > ### Comment · Reviewer_AbHb · 2023-08-14
> >
> > Wow, thank you! This is an incredible response and I have raised my score by 2 points.
> >
> > **Regarding proof of ALSH:** I am happy to see a proof of ALSH, though the $L_{\cos}$-Lipschitz condition effectively restricts the range of  $p_1$, $p_2$ for which the ALSH guarantee holds (because $d / dx \arccos(x) \to -\infty$ as $x \to 1$, so really high $p_2$ is a problem). Otherwise I was able to verify the proof. Your result is particularly nice because it shows how to set $M$ (which determines the time and space complexity of the hash).
> >
> > **Regarding general framework:** This is fantastic! My only question here is whether the ALSH-style $p_1, p_2$ guarantees can be extended to the general framework, perhaps by assuming some additional (smoothness?) conditions on $a(\mathbf{q} - \mathbf{x})$. This might let you get guarantees for hinge distance, ColBERT, box embeddings etc as specific instantiations (note that this is possibly out of scope, potentially something to consider in follow-up work if it is nontrivial / not easy).
> >
> > **Regarding experiments:** Thank you for conducting an evaluation with FAISS. This is a solid result and it increases my confidence in the method. It might be possible to use hinge distance for the voronoi clustering / quantization too, but this is likely beyond what a practitioner would be expected to do in adapting FAISS for their specific use case.
> >
> > **Regarding graph baselines:** I still think it is important to compare with a graph-based index (HNSW or one of the many follow-ups: DiskANN, SpeedANN, EFANNA - they are all variations of the same core algorithm, so you would only really need to compare against one). Even though FLORA > HNSW and FourierHashNet > FLORA, this *does not* imply that FourierHashNet > HNSW because the experiment setups are much different. FLORA is a semantic hashing method that attempts to learn the representation space (embeddings) alongside the hashing function - the advantage of FLORA is likely a better data representation and not a faster search (LSH loses to HNSW evaluations where the metric is not learnable). The FLORA evaluation also implements HNSW in a very nonstandard way (instead of a distance metric, they use a "semantic relevance" score defined by a neural network). HNSW will likely do much better for the hinge distance over pre-trained embeddings because this obeys the triangle inequality.
> >
> > I do recognize that it would be difficult to run this during the rebuttal period (due to time constraints, and also because `nmslib/Hnswlib` is hard to extend - though in this case, adding hinge distance is probably do-able just by inheriting from `SpaceInterface`). But it's important point to make. However, it wouldn't substantially change my review - I think your truncate-transform-sample framework is significant, even if it loses to HNSW, because of the many aforementioned applications of asymmetric LSH.
> >
> > **Regarding revision:** Please do your best to integrate the general framework / ALSH proofs into the next version of the paper. For readers coming from the hashing / LSH community, the framework will be the most interesting part of the paper.
> >
> > **Minor points / follow-up questions:**
> > - In the revision, it would be great to have a table of all the new similarities that are now LSH-able, to highlight the impact.
> > - It would also be nice (but non-essential) to see the empirical collision probability (take 100 FourierHashNet hashes and plot the average collision rate between points against the ground-truth hinge distance).
> > - For new experiment, is the wall-clock speedup against FAISS also ~2x? Hashing / graph algorithms are sometimes slower than cluster search, even though they perform fewer distance calculations, due to their memory access pattern and cache locality effects.
> >
> > To summarize, your response completely changed my view on this paper. All of the important points are addressed and I plan to argue in favor of acceptance. Great job!

---

> > > ### Author Response · Authors · 2023-08-20
> > >
> > > We once again extend our sincere gratitude to the reviewer for their insightful and appreciative feedback.
> > >
> > > > I do recognize that it would be difficult to run this during the rebuttal period (due to time constraints, and also because nmslib/Hnswlib is hard to extend - though in this case, adding hinge distance is probably do-able just by inheriting from SpaceInterface). But it's important point to make. However, it wouldn't substantially change my review - I think your truncate-transform-sample framework is significant, even if it loses to HNSW, because of the many aforementioned applications of asymmetric LSH.
> > >
> > >
> > > As per your suggestion, we extended the `SpaceInterface` class of `nmslib/Hnswlib`  to implement HNSW for hinge distance.
> > > In order to track the number of distance computations performed by HNSW during retrieval, we used a counter inside `fstdistfunc_`.
> > > We searched across different values of  `M`, `ef` and `ef_construction`, and tracked the number of distance computations against corresponding MAP values.
> > >
> > >
> > > |#calls to `fstdistfunc_`|  MAP |     Method     |
> > > |:----------------------:|:----:|:--------------:|
> > > |           839          | 0.17 |      HNSW      |
> > > |          1162          | 0.53 | FourierHashNet |
> > > |          1549          | 0.43 |      HNSW      |
> > > |          1668          | 0.51 |      HNSW      |
> > > |          1846          | 0.58 | FourierHashNet |
> > > |          2578          | 0.59 |      HNSW      |
> > > |          3529          | 0.71 |      HNSW      |
> > > |          3926          | 0.72 | FourierHashNet |
> > > |          4773          | 0.77 |      HNSW      |
> > > |          5347          | 0.74 | FourierHashNet |
> > > |          6694          | 0.81 |      HNSW      |
> > >
> > > The table (perhaps better viewed as a scatter) presents our study on MSWEB dataset with 10734 corpus items. We observe that FourierHashNet LSH has an edge over HNSW in the regime of fewer distance computations, with a MAP of 0.53 using 1162 distance computations. However, when allowed more distance computations, HNSW outperforms FourierHashNet with a MAP of 0.71 in 3528 computations, and a MAP of 0.77 in 4773 computations.
> > >
> > > (We count the number of pseudo-distance computations as a surrogate for real time, to avoid non-determinism in measurements and low-level implementation differences.  It was not possible to *exactly* equalize the number of distance computations performed by HNSW and FourierHashNet by tuning their respective hyperparameter.  HNSW has many performance-tuning parameters; we will present a more complete exploration of this space in the updated manuscript.)
> > >
> > > > In the revision, it would be great to have a table of all the new similarities that are now LSH-able, to highlight the impact.
> > >
> > > Thanks! We will make sure to do so.
> > >
> > >
> > > > It would also be nice (but non-essential) to see the empirical collision probability (take 100 FourierHashNet hashes and plot the average collision rate between points against the ground-truth hinge distance).
> > >
> > > | Hinge distance ⟶ | >1e+1 | [1e+1,1e+0] | [1e+0,1e-1] | <1e-2 |
> > > |:-------------------------:|:-----:|:-----------:|:-----------:|:-----:|
> > > |         #Buckets↓         |       |             |             |       |
> > > |            2^5            | 0.024 |     0.08    |     0.24    |  0.32 |
> > > |            2^7            | 0.005 |     0.04    |     0.15    |  0.21 |
> > > |            2^9            | 0.001 |     0.02    |     0.1     |  0.14 |
> > >
> > >
> > > We present the empirical collision probabilities for some randomly sampled embedding pairs. The embedding pairs are sampled such their hinge distances are at varying orders of magnitude. The columns of the table indicate the distances in decreasing order, while the rows indicate the number of buckets in the hash tables as dictated by the hashcode lengths.
> > >
> > > > For new experiment, is the wall-clock speedup against FAISS also ~2x? Hashing / graph algorithms are sometimes slower than cluster search, even though they perform fewer distance calculations, due to their memory access pattern and cache locality effects.
> > >
> > > We analyze the wall-clock times, for various values of number of comparisons (K), for both FAISS-IVF and FourierHashNet. For meaningful benchmarking, we implement FourierHashNet utilizing Falconn's C++ LSH APIs,  while comparing against FAISS-IVF's C++ code.
> > >
> > > Our study confirms that FourierHashNet's ~2x speedup against FAISS-IVF, earlier reported in terms of number of comparisons, also holds in terms of wall clock time. In the final draft, we will add a scatter plot for illustration.

---

### Official Review · Reviewer_2JDE · 2023-07-07

**Soundness:** 3 good
**Presentation:** 3 good
**Contribution:** 3 good
**Rating:** 7
**Confidence:** 3

**Summary:**

This paper studies a new search problem called vector dominance (or set containment), where the authors provide strong motivation from various real-world applications. They present a new approach named FourierHashNet along with a fresh asymmetric vector dominance distance to address this problem. Through extensive experimentation, the study confirms the effectiveness of the proposed distance measure and demonstrates the superior performance of FourierHashNet compared to established LSH and ALSH baselines.

**Strengths:**

- The authors study a new yet significant problem, which contains many well-motivated applications regarding text, image, and graph retrieval.

- They propose an effective distance measure customized for the problem they looked at.

- The FourierHashNet is interesting, and it is reasonable to combine learning to hash together with Fourier features.

- They conduct extensive experiments to confirm the effectiveness of FourierHashNet.

**Weaknesses:**

Overall, in my opinion, this is a well-written paper in various aspects.

**Questions:**

Please further proofread and polish the paper.

**Limitations:**

This work does not appear to have any negative social impact.

---

> ### Author Rebuttal · Authors · 2023-08-09
>
> The authors thank reviewer 2JDE for the positive feedback. We will undertake thorough proofreading and polishing for the final version of the manuscript.

---

### Author Rebuttal · Authors · 2023-08-09

# Truncate-transform-sample paradigm

### General Framework

Indeed, our algorithm can be generalized to include a wide variety of scoring functions, including Box embedding based volume scores, facility location scores used by ColBERT, etc.

Our framework extends to any shift-invariant scoring function of the form: $a(\pmb{q} -\pmb{x})$, which remains unchanged if $\pmb{q}$ and $\pmb{x}$ are shifted by the $\pmb{\delta}$. Assuming bounded values $||\pmb{q}|| _{\infty} \le q _{\max}$ and $||\pmb{x}|| _{\infty} \le x _{\max}$, a truncated function $sim(q,x) = s(\pmb{q} -\pmb{x}) = a(\pmb{q} -\pmb{x}) -a
_{\min}$, is active only within the specified bounds, and zero otherwise (In our case, $a = -d(q,x)$ and $a _{min} = -KT$).

This allows an absolutely convergent (because of trunctation) Fourier transformation:

\begin{align}
S(\pmb{\omega})=\frac{1}{(2\pi)^K} \int_{\pmb{t} \in \mathbb{R}^K}s(t) e^{-i \pmb{ \omega}^\top \pmb{t} } d \pmb{ t }
\end{align}
and the Inverse Fourier Transform of $s(\pmb{q} -\pmb{x})$ as:
\begin{align}
 s(\pmb{q} -\pmb{x}) & = \int_{\pmb{\omega} \in \mathbb{R}^K} S(\pmb{\omega})  e^{i \pmb{ \omega}^\top (\pmb{q} -\pmb{x}) } d \pmb{ \omega}
  = \int_{\pmb{\omega} \in \mathbb{R}^K} \pmb{S}_q(\pmb{\omega}) ^{\top} \pmb{S}_x(\pmb{\omega}) d\pmb{\omega}
\end{align}

Here,
\begin{align}
  \pmb{S}_q( \pmb{\omega}) {=} \Big[\text{Sign}(Re(S(\pmb{\omega}))\sqrt{|Re(S(\pmb{\omega}))|} \big[\cos(\pmb{\omega}^\top \pmb{q}), \sin(\pmb{\omega}^\top \pmb{q})\big],
\text{Sign}(Im(S(\pmb{\omega})) \sqrt{| Im(S(\pmb{\omega})|}\big[-\sin(\pmb{\omega}^\top \pmb{q}), \cos(\pmb{\omega}^\top \pmb{q}) \big]\Big]
\end{align}
and
\begin{align}
  \pmb{S}_x( \pmb{\omega}) {=} \Big[ \sqrt{|Re(S(\pmb{\omega}))|} \big[\cos(\pmb{\omega}^\top \pmb{q}), \sin(\pmb{\omega}^\top \pmb{q})\big],
  \sqrt{| Im(S(\pmb{\omega})|}\big[\cos(\pmb{\omega}^\top \pmb{q}), \sin(\pmb{\omega}^\top \pmb{q}) \big]\Big]
\end{align}

Above expressions are similar to Eq (7) in our paper, where they were defined for each component frequency $\omega_k$, thanks to the decomposability of the score functions as a sum of independent scores across dimensions $(s(\pmb{q}-\pmb{x}) = \sum  _{k=1} ^K s(q[k]-x[k]) )$. In contrast, here, we show that the setup can be extended to generic (shift invariant) scoring functions which need not be decomposable as a sum across dimensions.

However, we can define a similar distribution $p(\pmb{\omega})$ over the vector $\pmb{\omega}$ and obtain $s(\pmb{q} -\pmb{x})  = \mathbb{E} _{p(\pmb{\omega})}  [  \pmb{S}_q( \pmb{\omega})^{\top}   \pmb{S}_x( \pmb{\omega})/p(\pmb{\omega})]$.

### ColBERT

E.g., given a query $q = (q _1,.., q _m)$ and one corpus item $x  = (x_1,...,x_n)$, ColBERT computes the (facility location based) similarity scores between these two sets as
\begin{align}
\operatorname{sim}(q,x) =\sum _{i =1} ^m  \max _{j \in [n]} a(q _i,x _j)
\end{align}
If $a$ is a shift invariant score $a(\pmb{q} _i-\pmb{x} _j)$, say, inverse to Euclidean distance, then we can write its soft surrogate
\begin{align}
\operatorname{sim}(q,x) &=\frac{1}{\lambda} \sum _{i =1} ^m \log \left( \sum _{j\in [n]} \exp(\lambda a(\pmb{q} _i-\pmb{x} _j) )\right)
\end{align}

Note that the function $sim(q,x)$ is a shift invariant function of the form $s(\pmb{q}-\pmb{x})$.

### Box Embeddings

The box embedding based volume score can be expressed as a shift invariant score $a(\pmb{q} - \pmb{x})$. Here, the query and corpus items are expressed as boxes denoted by $(\pmb{z} _q, \pmb{Z} _q )$ and $(\pmb{z} _x, \pmb{Z} _x )$ respectively (the lower and upper corner coordinate vectors). The hard intersection between $q$ and $x$ is then the box $(\pmb{z, Z})$ where $\pmb{z} _{q,x}=  \max ( \pmb{z} _q, \pmb{z} _x, )$ and  $\pmb{Z} _{q,x} =  \min ( \pmb{Z} _q, \pmb{Z} _x, )$. Then the score between q,x is measured as:
\begin{align}
\operatorname{sim}(q,x) = \prod _{k = 1} ^K [\pmb{Z}
_{q,x}[k]- \pmb{z} _{q,x}[k]] _+  \hspace{3cm} ---(S)
\end{align}


We will show that there exists embedding $\pmb{q}$ and $\pmb{x}$ for which $sim (q,x) = a( \pmb{q} - \pmb{x})$. We first note that $\max(x,y) = x + (y-x) _+$ and $\min(x,y) =  y - (y-x) _+$. Using them, we have  $\pmb{z} _{q,x} =  \pmb{z} _{q} +( \pmb{z} _{x}-\pmb{z} _{q}) _+$ and $\pmb{Z} _{q,x} =  \pmb{Z} _{x} -( \pmb{Z} _{x}-\pmb{Z} _{q}) _+$. Thus, Eq. (S) is written as

\begin{align}
\operatorname{sim}(q,x) = \prod _{k = 1} ^K [  \pmb{Z} _{x} -\pmb{z} _{q} -( \pmb{z} _{x}-\pmb{z} _{q})
_+ -( \pmb{Z}
 _{x}-\pmb{Z}
 _{q}) _+ ]   _+ [k]
\end{align}

If we represent $\pmb{q} = [\pmb{z} _{q} , \pmb{z} _{q} ,\pmb{Z} _{q}  ]$ and $\pmb{x} = [\pmb{Z} _{x}, \pmb{z} _{x}, \pmb{Z}  _{x}   ]$, then we have
$\operatorname{sim}(q,x) = \prod _{k = 1} ^K [A  _1 ( \pmb{q}  -\pmb{x} ) -  [A _2 ( \pmb{q}  -\pmb{x} )  ] _+ -[A _3  ( \pmb{q}  -\pmb{x} )] _+] _+$ where $A _1 = -[\mathbb{I}, 0 ,0], A _2 = -[ 0, \mathbb{I} ,0]$ and $A _3= -[0,0,\mathbb{I}]$.

Thus $\operatorname{sim}(q,x)$ is shift invariant with respect to $\pmb{q}$ and $\pmb{x}$. Thus, we can extend our algorithm to box embedding setup.

# Citations

[Box and Order Embeddings]
T Chheda et al.  "Box Embeddings: An open-source library for representation learning using geometric structures". arXiv:2109.04997

[FLORA]
K Doan et al. "Asymmetric Hashing for Fast Ranking via Neural Network Measures". SIGIR 2023. arXiv:2211.00619

[NeuroMatch]
R Ying et al. "Neural Subgraph Matching" arXiv:2007.03092

[BERT-INT]
X Tang et al. “BERT-INT: A BERT-based Interaction Model For Knowledge Graph Alignment.”  IJCAI 2020.

[SNLI]
S Bowman et al. A large annotated corpus for learning natural language inference. EMNLP 2015.

[MNLI]
A Williams et al. A Broad-Coverage Challenge Corpus for Sentence Understanding through Inference. NAACL 2018.

---

### Decision · Program_Chairs · 2023-09-21

**Decision:**

Accept (spotlight)

**Comment:**

The paper presents a neat idea of LSH in Fourier domain for set containment search, a very important problem in information retrieval when the query and data have asymmetrically different lengths or norms. Two reviewers were very excited about the papers and one raised concerns. The rebuttal was able to convince the concerned reviewer and in the end all reviewers unanimously agree that this paper is above the bar for acceptance.